# Cell culture-based profiling across mammals reveals DNA repair and metabolism as determinants of species longevity

Siming Ma[1], Akhil Upneja[1], Andrzej Galecki[2,3,4], Yi-Miau Tsai[2,3], Charles F Burant[5], Sasha Raskind[5], Quanwei Zhang[6], Zhengdong D Zhang[6], Andrei Seluanov[7], Vera Gorbunova[7], Clary B Clish[8], Richard A Miller[2,3], Vadim N Gladyshev[1]*

[1]Division of Genetics, Department of Medicine, Brigham and Women's Hospital, Harvard Medical School, Boston, United States; [2]Department of Pathology, University of Michigan Medical School, Ann Arbor, United States; [3]Geriatrics Center, University of Michigan Medical School, Ann Arbor, United States; [4]Department of Biostatistics, School of Public Health, University of Michigan, Ann Arbor, United States; [5]Department of Internal Medicine, University of Michigan Medical School, Ann Arbor, United States; [6]Department of Genetics, Albert Einstein College of Medicine, Bronx, United States; [7]Department of Biology, University of Rochester, Rochester, United States; [8]Broad Institute, Cambridge, United States

*For correspondence:
vgladyshev@rics.bwh.harvard.edu

Competing interests: The authors declare that no competing interests exist.

**Abstract** Mammalian lifespan differs by >100 fold, but the mechanisms associated with such longevity differences are not understood. Here, we conducted a study on primary skin fibroblasts isolated from 16 species of mammals and maintained under identical cell culture conditions. We developed a pipeline for obtaining species-specific ortholog sequences, profiled gene expression by RNA-seq and small molecules by metabolite profiling, and identified genes and metabolites correlating with species longevity. Cells from longer lived species up-regulated genes involved in DNA repair and glucose metabolism, down-regulated proteolysis and protein transport, and showed high levels of amino acids but low levels of lysophosphatidylcholine and lysophosphatidylethanolamine. The amino acid patterns were recapitulated by further analyses of primate and bird fibroblasts. The study suggests that fibroblast profiling captures differences in longevity across mammals at the level of global gene expression and metabolite levels and reveals pathways that define these differences.

## Introduction

The maximum lifespan of mammalian species differs by more than 100-fold, ranging from ~2 years in shrews to >200 years in bowhead whales (*Tacutu et al., 2013*). While it has long been observed that maximum lifespan tends to correlate positively with body mass and time to maturity, but negatively with growth rate, mass-specific metabolic rate, and number of offspring (*Peters, 1986*; *Sacher, 1959*; *Western, 1979*), the underlying molecular basis is only starting to be understood.

One way to study the control of longevity is to identify the genes, pathways, and interventions capable of extending lifespan or delaying aging phenotypes in experimental animals. Studies using model organisms have uncovered several important conditions, such as knockout of insulin-like growth factor 1 (IGF-1) receptor (*Friedman and Johnson, 1988*; *Holzenberger et al., 2003*;

*Tatar et al., 2001*), inhibition of mechanistic target of rapamycin (mTOR) (*Harrison et al., 2009*; *Kenyon, 2010*; *Miller et al., 2014*), mutation in growth hormone (GH) receptor (*Coschigano et al., 2000*), ablation of anterior pituitary (e.g. Snell dwarf mice) (*Flurkey et al., 2002*), augmentation of proteins of the sirtuin family (*Chang and Guarente, 2013*; *Gomes et al., 2013*; *Mouchiroud et al., 2013*; *Wood et al., 2004*), and restriction of dietary intake (*Guarente and Kenyon, 2000*; *Heilbronn and Ravussin, 2003*; *McCay et al., 1935*; *Weindruch et al., 1986*). While many of these genes and pathways have been verified in yeast, flies, worms, and mice, the comparisons largely involve treatment and control groups of the same species, and the extent to which they explain the longevity variations across different species is unclear. For example, do the long-lived species have metabolic profiles resembling calorie restriction? Do they suppress IGF-1 or growth hormone signaling compared with the shorter-lived species? More generally, how do the evolutionary strategies of longevity relate to the experimental strategies that extend lifespan in model organisms?

To address these questions, a popular approach has been to compare exceptionally long-lived species with closely related species of common lifespan and identify the features associated with exceptional longevity. Examples include the amino acid changes in Uncoupling Protein 1 (UCP1) and production of high-molecular-mass hyaluronan in the naked mole rat (*Kim et al., 2011*; *Tian et al., 2013*); unique sequence changes in IGF1 and GH receptors in Brandt's bat (*Seim et al., 2013*); gene gain and loss associated with DNA repair, cell-cycle regulation, and cancer, as well as alteration in insulin signaling in the bowhead whale (*Keane et al., 2015*; *Seim et al., 2014*); and duplication of the p53 gene in elephants (*Abegglen et al., 2015*). Again, it is important to ascertain whether these mechanisms are unique characteristics of specific exceptionally long-lived species, or whether they can also help account for the general lifespan variation (*Martin, 1988*; *Partridge and Gems, 2002*).

An extension of this approach has been cross-species analyses in a larger scale. For example, several biochemical studies across multiple mammalian and bird species identified some features correlating with species lifespan. Longevity of fibroblasts and erythrocytes in vitro (*Röhme, 1981*), poly (ADP-ribose) polymerase activity (*Grube and Bürkle, 1992*), and rate of DNA repair (*Cortopassi and Wang, 1996*) were found to be positively correlated with longevity, whereas mitochondrial membrane and liver fatty acid peroxidizability index (*Pamplona et al., 1998*, *2000*), rate of telomere shortening (*Haussmann et al., 2003*), and oxidative damage to DNA and mitochondrial DNA (*Adelman et al., 1988*; *Barja and Herrero, 2000*) showed negative correlation. The advent of high-throughput RNA sequencing (RNAseq) and mass spectrometry technologies has enabled the quantification of whole transcriptomes (*Fushan et al., 2015*), metabolomes (*Ma et al., 2015b*), and ionomes (*Ma et al., 2015a*), across multiple species and organs. These studies revealed the complex transcriptomic and metabolic landscape across different organs and species, as well as some overlaps with the changes observed in the long-lived mutants created in laboratory (*Ma et al., 2015b*).

While molecular profiling of mammals at the level of tissues may better represent the underlying biology, profiling in cell culture represents more defined experimental conditions and allows further manipulation to alter the identified molecular phenotypes. In this study, we examined the transcriptomes and metabolomes of primary skin fibroblasts across 16 species of mammals, to identify the molecular patterns associated with species longevity. We report that the genes involved in DNA repair and glucose metabolism were up-regulated in the longer lived species, whereas proteolysis and protein translocation activities were suppressed. The longer lived species also had lower levels of lysophosphatidylcholine and lysophosphatidylethanolamine and higher levels of amino acids; and the latter finding was validated in an independent dataset of bird and primate fibroblasts. Thus, molecular insights into longevity may indeed come from defined cell culture systems in mammals.

## Results

### Gene expression by RNA sequencing

To identify the molecular signatures associating with the differences in longevity, we obtained primary, sun-protected abdominal skin fibroblasts from 13 species of rodents, two species of bats and one species of shrew, representing a wide range of maximum lifespan (ML; from 2.2 years in shrew to 34.0 years in little brown bat) and adult weight (AW; from 10 g in little brown bat to 20 kg in beaver) (*Figure 1*, *Figure 1—source data 1A*). Female time to maturity (FTM) and the body mass adjusted residuals (i.e. MLres and FTMres) were included as additional longevity trait (*Figure 1*,

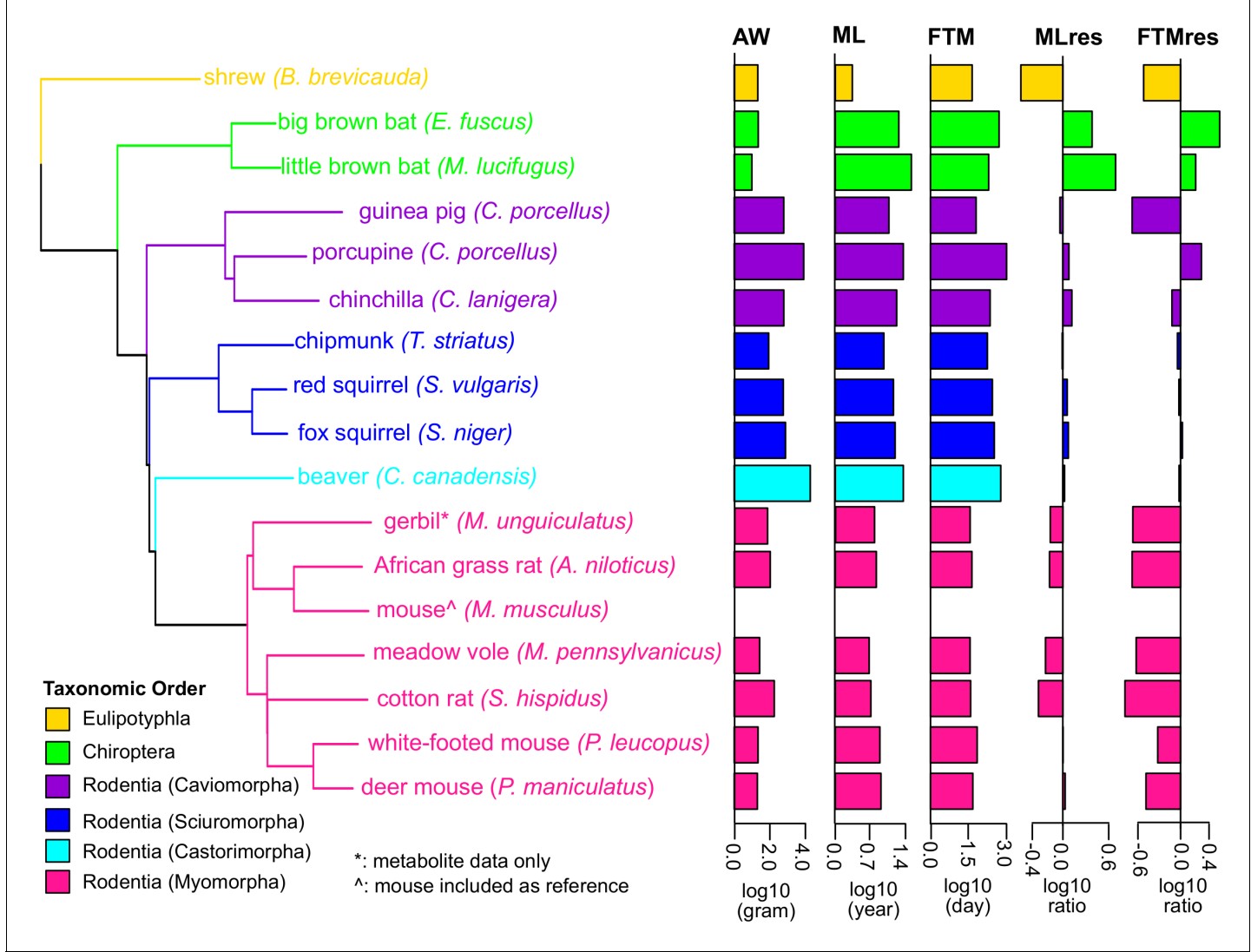

**Figure 1.** Phylogenetic relationship among species used in the study. The tree was constructed using Neighbor-Joining method based on nucleotide sequences. Shrew was used as the out-group. Gerbil was collected for metabolite data only and mouse was included as reference. The species are colored by taxonomic order. Adult Weight (AW), Maximum Lifespan (ML), Female Time to Maturity (FTM), Maximum Lifespan Residual (MLres), and Female Time to Maturity Residual (FTMres) of these species are displayed in log10 scale.

The following source data is available for figure 1:

**Source data 1.** Species and samples used in the current study.

Materials and methods). We profiled gene expression by RNAseq on 28 samples representing 15 species (except gerbil) (*Figure 1—source data 1B*). Only five of these species had publicly available genomes; this posed a challenge as reliable reference sequences were crucial for accurate RNAseq read alignment and read counting. The gene orthology information was also limited or unavailable for the less common species. To address these issues, we developed a pipeline to obtain species-specific ortholog sets (*Figure 2A*, Materials and methods). We defined a set of mouse reference sequences based on Ensembl and then performed de novo transcriptome assembly for each species. BLAST was used to find reciprocal best hits between the assembled transcriptome (and published genome, if available) and the mouse reference (*Altschul et al., 1997*; *Camacho et al., 2009*; *Tatusov et al., 1997*). The reciprocal best hits were then trimmed down to open reading frame and

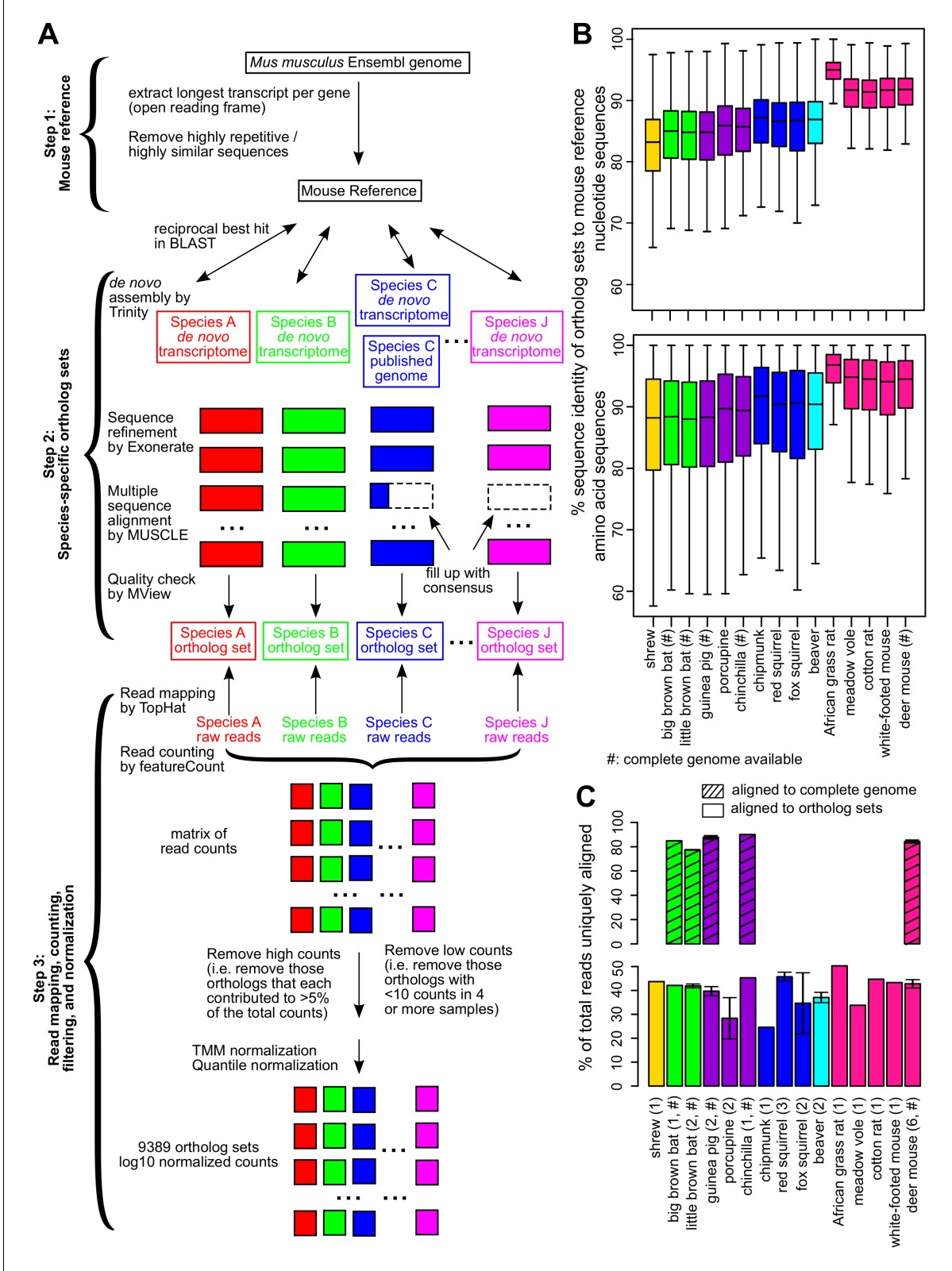

**Figure 2.** Cross-species analysis of gene expression in cultured skin fibroblasts. (**A**) Pipeline to obtain the species-specific ortholog sets and expression values. See Materials and methods or a more detailed description of the methodology. (**B**) Sequence identity of ortholog sets compared to mouse. Nucleotide and amino acid sequence identity of the ortholog sets in each species was compared to mouse reference (mouse was set as 100%). The ortholog sequences were based on de novo assembled transcriptomes, as well as NCBI genomes (if available; indicated by '#'). The box plot shows the

*Figure 2 continued on next page*

*Figure 2 continued*

distribution across the 9389 gene orthologs, with the central bars indicating median values. (C) Read alignment rates for mapping to complete genomes and ortholog sets. Percent of total reads that could be uniquely aligned to the complete genomes (if available, indicated by '#'; shaded bars) or to the ortholog sets are shown. Error bars refer to standard error of mean. Number of samples (biological and technical replicates) per species is indicated in parenthesis.

The following figure supplement is available for figure 2:

**Figure supplement 1.** Quality assessment of orthologs.

the quality of the ortholog sets was assessed by multiple sequence alignment (Materials and methods).

The median nucleotide sequence identity for our ortholog sets with respect to mouse ranged from 83.2% (shrew) to 95.0% (African grass rat), and protein sequence identity from 88.0% (little brown bat) to 96.8% (African grass rat) (*Figure 2B*), consistent with the evolutionary distance of the species to mouse. The read alignment rates were also largely consistent across samples (*Figure 2C*). For a number of sequences with poor coverage, the consensus sequences of closely related species were used instead, but this did not significantly affect the results (*Figure 2—figure supplement 1*). After data filtering and normalization (Materials and methods), the expression of 9389 gene orthologs was reliably detected across the 28 samples (*Supplementary file 1*). For those species with publicly available genomes, ~10,000–11,000 genes could be reliably detected and the read counts also showed strong agreement (Pearson correlation coefficient 0.95–0.98 for log10 counts; *Figure 1—source data 1C*).

## Gene expression patterns in fibroblasts follow phylogeny

To assess the gene expression patterns across the species, we performed Principal Component Analysis and projected the data on the first three Principal Components (*Figure 3A*). The samples segregated predominantly by their taxonomic relationship. For example, the species belonging to the sub-orders Sciuromorpha (chipmunk, red squirrel, and fox squirrel), Hystricomorpha (guinea pig, porcupine, and chinchilla), and Myomorpha (African grass rat, meadow vole, cotton rat, white-footed mouse, and deer mouse) separated clearly from one another (*Figure 3A*). The topology of the expression phylogram was also similar to the tree based on nucleotide sequences (*Figure 3B*), suggesting the expression patterns are influenced by phylogeny. In addition, the biological and technical replicates of the respective species clustered together, confirming that the within-species variation was generally smaller than the cross-species variation (*Ma et al., 2015b*).

## Expression of many genes correlates with longevity traits

To identify the genes with significant correlation to longevity, we performed regression by generalized least squares between the gene expression values and AW, as well as the four longevity traits (ML, FTM, MLres, and FTMres). The phylogenetic relationship of the species was incorporated in the variance-covariance matrix, and four different trait evolutionary models were tested to select the best models based on maximum likelihood (Materials and methods) (*Lavin et al., 2008*; *Ma et al., 2015b*). A two-step verification procedure was applied to assess robustness of the results (*Ma et al., 2015b*). Briefly, the potential outlier point was first identified and excluded to improve the regression fit (the regression slope p value was reported as 'p value.robust'). Regression was then repeated by excluding each species, one at a time, to report the maximal (i.e. least significant) p value ('p value.max'), to ensure the overall relationship did not depend on any single species.

We qualified as top hits those genes meeting both criteria of p value.robust < 0.01 (~11% FDR) and p value.max < 0.05. The numbers of top hits were 675 for AW, 812 for ML, 830 for FTM, 508 for MLres, and 793 for FTMres, with roughly equal proportions in positive and negative correlations (*Table 1—source data 1A–F*) and some overlap among the four longevity traits (*Figure 3C*). For most of the top hits, the directions of correlation were consistent across the four longevity traits (even for those that failed to reach statistical significance), suggesting there was a core set of longevity-associated genes and the minor inaccuracy in the reported lifespan data was unlikely to affect the overall results. On the other hand, the overlap with the hits identified by AW was much smaller

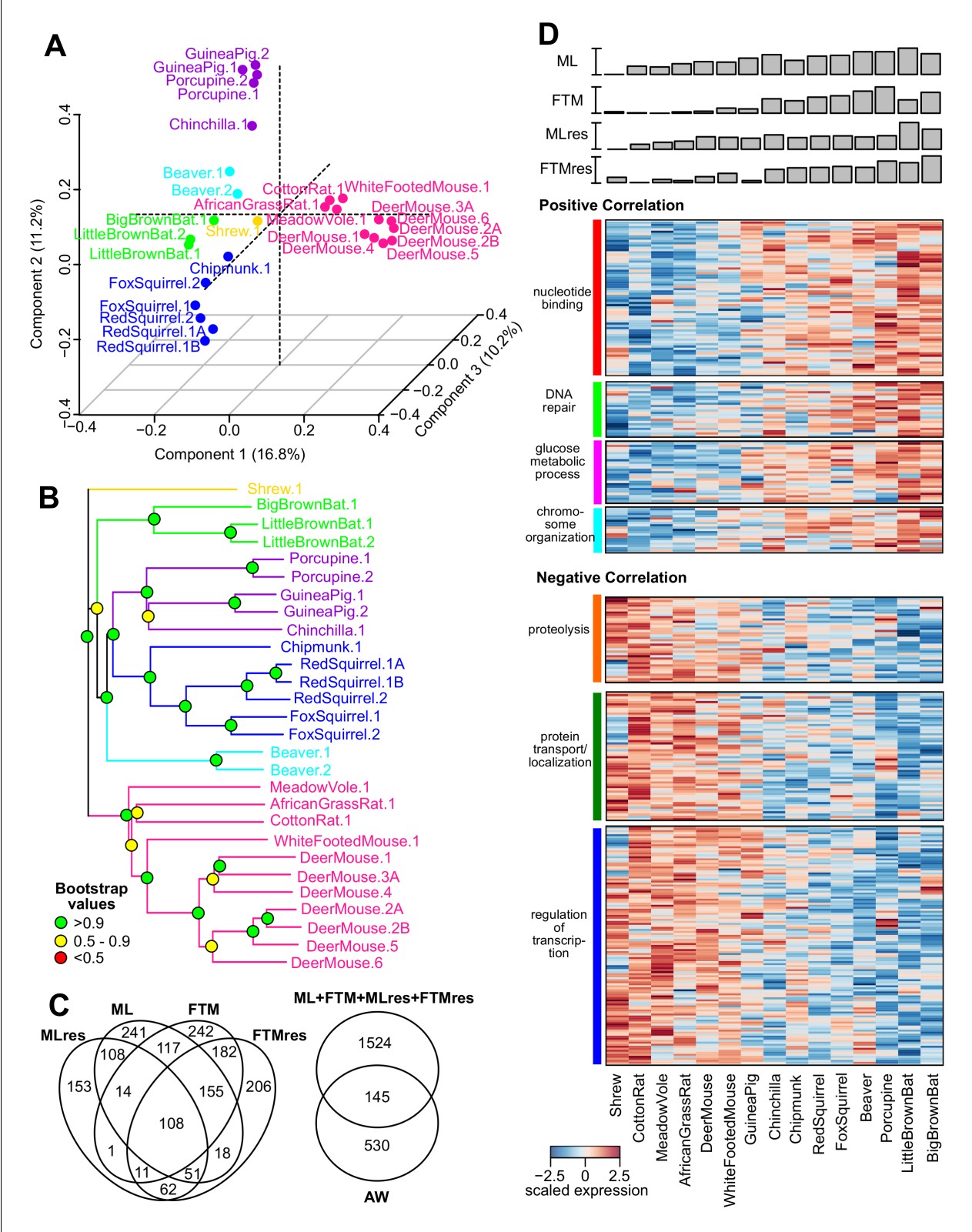

**Figure 3.** Gene expression variation and correlation with longevity. (**A**) Projection of the first three Principal Components (PCs) in Principal Component Analysis. Values in parenthesis indicate percentage of variance explained by each of the PCs. Points are colored by taxonomic order (same color scheme as in *Figure 1*) (**B**) Gene expression phylogram. Color of the nodes indicates the result of 1000 times bootstrap. (**C**) Overlap of genes associating with Adult Weight and indicated longevity traits. AW: Adult Weight; ML: Maximum Lifespan; FTM: Female Time to Maturity; MLres:

*Figure 3 continued on next page*

*Figure 3 continued*

Maximum Lifespan Residual; FTMres: Female Time to Maturity Residual. (**D**) Heat map showing expression patterns of the top enrichment pathways. Species are arranged in the order of increasing longevity (the four longevity traits are scaled between 0 and 1).

The following figure supplement is available for figure 3:

**Figure supplement 1.** Interaction network among the top hits in (**A**) positive and (**B**) negative correlation with longevity.

(*Figure 3C*), indicating the observed correlations were not driven mainly by body mass differences. For the 827 top hits supported by two or more longevity traits, we performed pathway enrichment analysis using DAVID (*Table 1*, *Table 1—source data 1G–H*) (*Huang da et al., 2009a*, *2009b*) based on mouse pathways.

## Genes showing positive correlation with lifespan

The top pathways for the genes with positive correlation included 'nucleotide binding' (15% of the genes with positive correlation to longevity), 'DNA repair' (4%), 'glucose metabolic process' (4%), and 'chromosome organization' (4%) (*Table 1*, *Figure 3D*). The 'DNA repair' genes included those in DNA mismatch repair (*Msh6*, *Pms2*), nonhomologous end joining and possibly other repair pathways (*Pnkp*), nucleotide excision repair and DNA double-strand break repair (*Ercc1*), Fanconi anemia-associated DNA damage response network (*C17orf70*, *Fancg*), and protection of telomeres (*Rif1*, *Terf1*, *Tinf2*). The products of checkpoint kinase *Chek1* and anaphase promoting complex substrate *Pttg1* were regulators of cell cycle.

Among the other genes, *Hif1a* encodes the alpha subunit of hypoxia-inducible factor 1 (HIF-1), a key transcription factor in mediating the metabolic responses to hypoxia, whereas *Prdx3* encodes mitochondrial peroxiredoxin that regulates redox homeostasis. In particular, *Pnkp* (*Figure 4A*), *Prdx3*, and *Rif1* reached statistical significance in all four longevity traits (*Table 1—source data 1F*). Consistent with the findings, over-expression of *hif-1* in *C. elegans* was shown to promote longevity (*Zhang et al., 2009*), whereas deletion of *rif1* and *msh6* in yeast (*Austriaco and Guarente, 1997*; *Laschober et al., 2010*), knockout of *prdx3* in *C. elegans* (*Ha et al., 2006*), and disruption of *Ercc1* in mouse (*Weeda et al., 1997*) were all detrimental and led to decreased lifespan. Several previous studies also suggested that long-lived species generally have enhanced DNA repair capacity (*Cortopassi and Wang, 1996*), higher poly (ADP-ribose) polymerase activity (*Grube and Bürkle, 1992*), up-regulation of genes in base-excision repair and superoxide metabolic process (*Fushan et al., 2015*), as well as reduced free-radical production (*Perez-Campo et al., 1998*), reduced oxidant generation (*Sohal et al., 1995*), and less oxidative damage to nuclear DNA (*Adelman et al., 1988*) and mitochondrial DNA (*Barja and Herrero, 2000*), although the degree of contribution toward the observed differences in lifespan varied and might be affected by several confounding effects (*Debrabant et al., 2014*; *Montgomery et al., 2012*; *Promislow, 1994*).

'Glucose metabolic process' included the gene products of hexokinase (*Hk1*), glucose phosphate isomerase (*Gpi1*), triose phosphate isomerase (*Tpi1*), phosphofructose kinase (*Pfkp*), and pyruvate dehydrogenase kinase (*Pdk1*), which are involved in glycolysis/gluconeogenesis. The glucan branching enzyme (encoded by *Gbe1*) and several phosphorylase kinases (encoded by *Phka2*, *Phkb*, *Phkg2*) regulate the metabolism of glycogen. In addition, the genes coding for NAD synthetase (*Nadsyn1*), which is involved in converting nicotinate adenine dinucleotide (NaAD) to nicotinamide adenine dinucleotide (NAD), also showed positive correlation with all four longevity traits (*Figure 4B*). Previously, it was observed that $NAD^+$ levels declined with age and affected SIRT1 functions, whereas supplementation with $NAD^+$ precursors reversed the aging phenotypes in mouse muscle (*Gomes et al., 2013*), and overexpression of SIRT1 in mouse brain could protect against aging-dependent circadian changes (*Chang and Guarente, 2013*). Calorie restriction also increases the $NAD^+$/NADH ratio in yeast (*Lin et al., 2004*). As our study did not directly quantify the $NAD^+$/NADH ratio, it remains to be seen if the high *Nadsyn1* expression in the fibroblasts of the long-lived species affects these metabolites.

**Table 1.** Pathway enrichment analysis of genes with significant correlation with the longevity traits. The genes were supported by at least two longevity traits (p value.robust < 0.01 and p value.max < 0.05). Pathway enrichment was performed using DAVID. The percentages of positive or negative correlating genes belonging to each pathway were indicated in parentheses. Only selected pathways are shown here. GO (BP): Gene Ontology (Biological Process). GO (BP): Gene Ontology (Molecular Functions). SP/PIR: SwissProt and Protein Information Resource. See **Table 1—source data 1** for more details.

| Annotation cluster | Enriched terms and genes | No. of genes | p Value |
|---|---|---|---|
| Positive Correlation Cluster No. 1 (15%) | GO (MF): adenyl nucleotide binding | 50 | $5.25 \times 10^{-3}$ |
| | GO (MF): nucleotide binding | 64 | $1.21 \times 10^{-2}$ |
| | Acly, Atad2, Atp2b4, Cdk2, Cdk20, Chd7, Chek1, Chkb, Cpsf7, D2hgdh, Dgkq, Dhx58, Dock6, Ero1lb, Etnk1, Fastkd5, Fn3krp, Gnai1, Guk1, Hk1, Hmgcr, Hnrnpd, Hyou1, Insr, Madd, Map4k5, Mastl, Mlkl, Mov10, Msh6, Mx2, Nadsyn1, Oplah, Pdk1, Pfkp, Phka2, Phkg2, Pkmyt1, Pms2, Pnkp, Ppp2r4, Prkar1b, Qrsl1, Rbm10, Rbm15b, Rbm38, Rhot2, Rnasel, Rps6ka2, Sacs, Sirt3, Slirp, Smarca1, Smarca5, Srsf9, Stk19, Stk36, Tbrg4, Tesk2, Thnsl1, Tia1, Top3a, Trpm4, Ttf2, Tyk2, Vps4a, Ythdc2 | | |
| Positive Correlation Cluster No. 2 (4%) | SP/PIR: DNA damage | 14 | $1.16 \times 10^{-3}$ |
| | SP/PIR: DNA repair | 12 | $4.25 \times 10^{-3}$ |
| | GO (BP): cellular response to stress | 16 | $1.01 \times 10^{-1}$ |
| | Bnip3, C17orf70, Chek1, Dtx3l, Ercc1, Errfi1, Fancg, Hif1a, Mapkbp1, Msh6, Myd88, Pms2, Pnkp, Prdx3, Prpf19, Pttg1, Rad51b, Rif1, Rnaseh1, Slx4, Tdp2, Terf1, Tinf2, Top3a, Wrap53 | | |
| Positive Correlation Cluster No. 4/5 (4%) | GO (BP): glucose metabolic process | 11 | $1.22 \times 10^{-3}$ |
| | GO (BP): hexose metabolic process | 11 | $5.68 \times 10^{-3}$ |
| | GO (BP): generation of precursor metabolites and energy | 15 | $4.59 \times 10^{-3}$ |
| | Aldh5a1, Atp2b4, Atp6v0d1, Atp6v0e2, Ero1lb, Fads1, Gbe1, Gpi1, Hexa, Hk1, Insr, Ndst1, Ndufa8, Pdk1, Pfkp, Pgp, Phka2, Phkb, Phkg2, Prkar1b, Sdhaf3, Tmx4, Tpi1, Trpm4, Tsc2 | | |
| Positive Correlation Cluster No. 6 (4%) | SP/PIR: chromatin regulator | 11 | $1.61 \times 10^{-2}$ |
| | GO (BP): chromosome organization | 17 | $2.22 \times 10^{-2}$ |
| | Bnip3, Cenph, Chd7, Dtx3l, Ercc1, H2afv, Hdac2, Jade1, Kdm5d, Kmt2c, Pttg1, Rcor1, Rrp8, Smarca1, Smarca5, Smyd3, Terf1, Tinf2, Wdr5, Wrap53 | | |
| Negative Correlation Cluster No. 1 (9%) | GO (BP): modification-dependent protein catabolic process | 27 | $2.39 \times 10^{-4}$ |
| | SP/PIR: ubiquitin conjugation pathway | 26 | $3.35 \times 10^{-4}$ |
| | GO (BP): proteolysis | 36 | $1.09 \times 10^{-2}$ |
| | Adamts2, Agtpbp1, Anapc4, Atg10, Atg4a, Atg7, Btbd1, Ctsl, Ctsz, Dcaf10, Dda1, Dpp8, Fbxl17, Fbxl20, Fbxo18, Fbxw2, Kcmf1, Map1lc3b, Med8, Mmp2, Mycbp2, Oma1, Pcsk5, Pgpep1, Pmepa1, Ppp2r5c, Rad18, Rfwd2, Rnf14, Rnf2, Rnf6, Sumo3, Tpp2, Ube2b, Ube2v1, Ufm1, Vhl | | |
| Negative Correlation Cluster No. 2 (9%) | GO (BP): protein localization | 38 | $4.67 \times 10^{-5}$ |
| | GO (BP): protein transport | 34 | $7.99 \times 10^{-5}$ |
| | Agap1, Akap7, Ap3d1, Atg10, Atg4a, Atg7, Bax, Cav1, Clpx, Cnih1, Col4a3bp, Cry2, Dirc2, Ergic2, Fdx1, Fkbp15, Gabarapl2, Gdi2, Gm10273, Golt1b, Hspa9, Ift46, Ipo4, Kif1bp, Kpna4, Laptm4a, Lrp4, mt-Nd4, Mtch1, Ndel1, Ndufb11, Necap1, Ppp3ca, Rab18, Rab2a, Rab6a, Rhot1, Sar1a, Sec22a, Sec31a, Sec62, Slc25a12, Slc29a1, Slc33a1, Slc35a4, Snx12, Snx13, Stx17, Timm8a1, Tomm6, Trappc6b, Trp53, Tsg101, Vps36, Vps53, Ywhag | | |

*Table 1 continued*

| Annotation cluster | Enriched terms and genes | No. of genes | p Value |
|---|---|---|---|
| Negative Correlation Cluster No. 3 (18%) | GO (BP): regulation of transcription | 74 | $1.62 \times 10^{-5}$ |
| | SP/PIR: transcription regulation | 55 | $1.04 \times 10^{-3}$ |
| | Actl6a, Agtpbp1, Ak6, Anp32a, Anp32e, Atf6b, Bckdha, Bmi1, Ccdc59, Cd3eap, Cdc5l, Cggbp1, Clk2, Cnbp, Cops7a, Crtc3, Cry2, Csrp2, Ebna1bp2, Ehmt2, Elk4, Ergic2, Fbxo18, Fip1l1, Fosb, Foxo3, Gatad2b, Gid8, Gmcl1, Gtf2h1, Gtf2h2, Gtf2h5, Harbi1, Hlx, Hmga1-rs1, Hnrnpab, Hnrnpf, Ift57, Ing2, Ints4, Ipo4, Jund, Klf11, Klf2, Klf4, Klf9, Kpna4, Mafb, Mapk1, Mdm4, Med8, Med16, Med17, Med31, Med8, Mef2a, Mettl8, Mmp2, Mnt, Morf4l2, Mta1, Mtdh, Mxd1, Mycbp2, Nabp2, Ncor2, Neo1, Nfe2l2, Nr1d2, Papd4, Parp2, Phf12, Phlpp1, Pkig, Pomp, Pop5, Ppp1r8, Ppp2r5c, Ppp3ca, Ptbp1, R3hdm4, Rab18, Rad18, Rbbp4, Rfwd2, Rnf14, Rnf2, Rnf6, Rps6ka4, Rrs1, Sap30l, Sav1, Scoc, Sfmbt1, Sin3b, Snrk, Sqstm1, Srpk2, Ssbp1, Tep1, Tgfbr3, Trim35, Trip6, Trp53, Tsg101, Ube2b, Ube2v1, Ubtf, Ufm1, Vhl, Vps36, Wiz, Xrcc5, Yeats4, Zbtb14, Zfp414, Zfp637, Zfp655, Zfp710, Zfp821 | | |

**Source data 1.** Phylogenetic regression of gene expression against longevity traits. Regression against (**A**) Adult Weight; (**B**) Maximum Lifespan; (**C**) Female Time to Maturity; (**D**) Maximum Lifespan Residual; and (**E**) Female Time to Maturity Residual. 'coef.all', 'p value.all', and 'q value.all' refer to the regression slope, p value, and FDR-adjusted q value using all the species. 'p value.robust' and 'q value.robust' refer to the statistics after removing the potential outlier point. 'p value.max' and 'q value.max' refer to the maximal (least significant) regression p value and q value when each one of the species was left out, one at a time. Only genes with p value.robust<0.01 and p value.max<0.05 are shown. (**F**) Top hits identified by two or more longevity traits. The p value.robust against each of the four longevity traits (ML, FTM, MLres, and FTMres) as well as adult weight (AW) are shown. These genes were the input for pathway enrichment analysis. Pathway enrichment analysis of genes showing (**G**) positive and (**H**) negative correlation with longevity traits. Enrichment was performed using DAVID with default settings. Only the top 10 clusters are shown. (**I**) System level analyses of gene functions. The numbers of shared genes between longevity associated genes (either positively or negatively or both) and human aging genes, essential genes, transcription factor genes, and housekeeping genes are shown. The enrichment p value was calculated by Fisher's exact test with two different background gene sets.

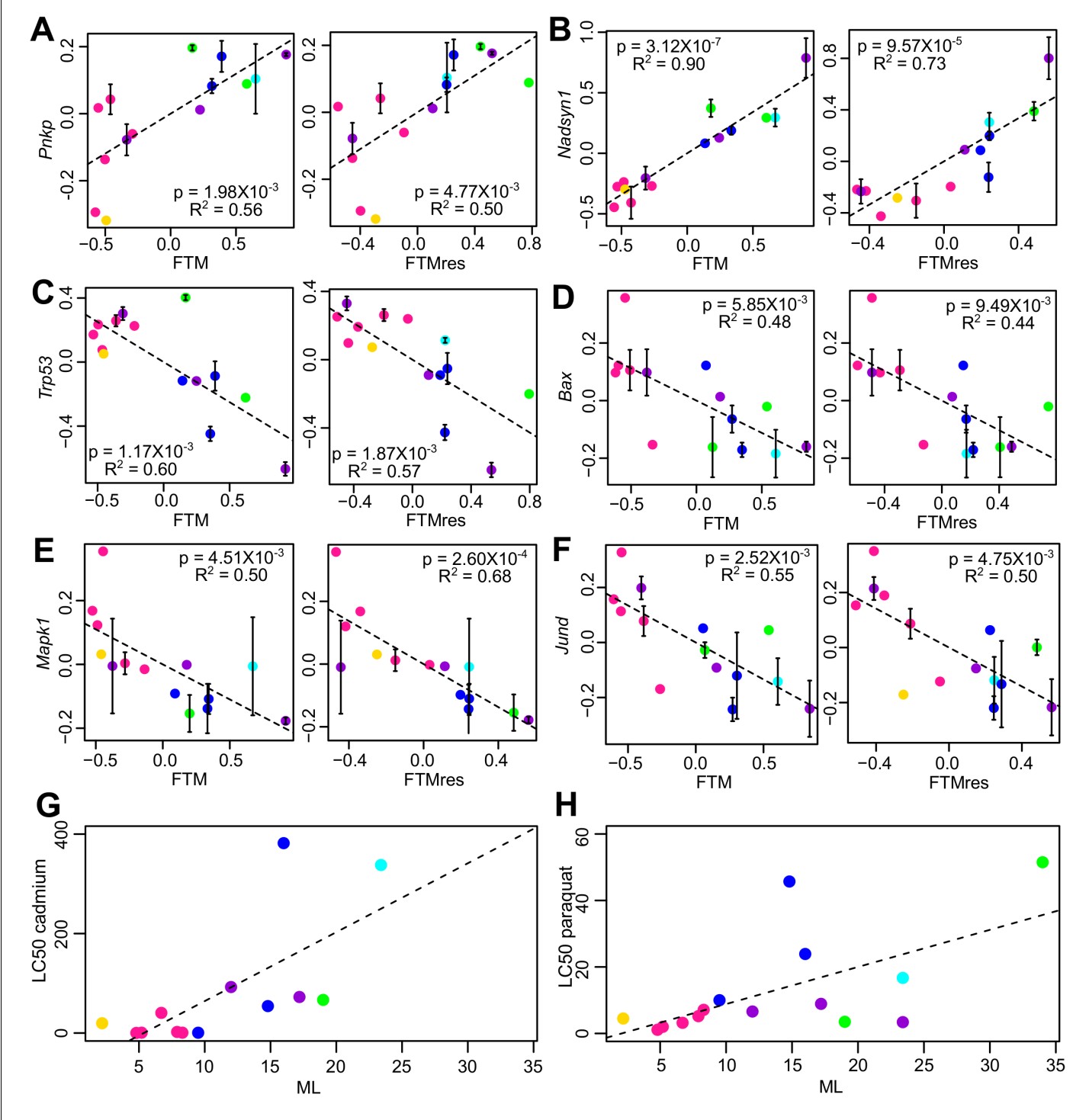

**Figure 4.** Selected genes and stress resistance conditions with significant correlation to longevity. (A) *Pnkp* and (B) *Nadsyn1* show positive correlation with the longevity traits. (C) *Trp53*, (D) *Bax*, (E) *Mapk1*, and (F) *Jund* show negative correlation with the longevity traits. In each plot, the gene expression values (vertical axis) and the longevity traits (horizontal axis; FTM: Female Time to Maturity; FTMres: Female Time to Maturity Residual) are centered at 0 on log10 scale and then transformed by the best-fit variance-covariance matrix under phylogenetic regression (i.e. to remove the phylogenetic relationship). The potential outlier point has been removed and the remaining points are shown on the plot and colored by taxonomic group (same color scheme as in *Figure 1*). The regression slope p value (i.e. p value.robust) and $R^2$ value are indicated. Error bars indicate standard error of mean. Resistance to (G) cadmium and (H) paraquat treatments. In each plot, the lethal dose (LD50) values (vertical axis) and the longevity traits

*Figure 4 continued on next page*

*Figure 4 continued*

(horizontal axis; ML: Maximum Lifespan) are plotted on ordinary scale (without log transformation). The regression slope p values are $9.16 \times 10^{-3}$ and $1.39 \times 10^{-2}$, respectively.

## Genes showing negative correlation with lifespan

With regard to the top hits showing negative correlation, the major enriched pathways included 'proteolysis' (9% of the genes with negative correlation to longevity), 'protein transport/localization' (9%), and 'regulation of transcription' (18%) (*Table 1*, *Figure 3D*). For 'proteolysis', we observed relatively low expression of the genes coding for E2 ubiquitin-conjugating enzyme (*Ube2b*, *Ube2v1*), E3 ubiquitin-protein ligase (*Rad18*, *Mycbp2*), ubiquitin-like modifier (*Sumo3*, *Ufm1*), as well as several proteins containing RING finger domain (*Rnf2*, *Rnf6*, *Rnf14*, *Rfwd2*) or F-box domain (*Fbxl17*, *Fbxl20*, *Fbxo18*, *Fbxw2*), both of which are known to be involved in the ubiquitination pathway. Also, low expression was observed for the genes encoding autophagy related proteins (*Atg4a*, *Atg7*, *Atg10*) and lysosomal cysteine proteinases (*Ctsl*, *Ctsz*). The genes implicated in 'protein transport/localization' included several vesicle trafficking proteins (*Sec22a*, *Sec31a*, *Sec62*, *Golt1b*), mitochondrial membrane translocases (*Timm8a1*, *Tomm6*), and nuclear transport receptors (*Ipo4*, *Kpna4*). As for 'regulation of transcription', we observed down-regulation of the genes coding for mediator complex subunits (*Med8*, *Med16*, *Med17*, *Med31*), zinc finger proteins (*Zfp414*, *Zfp655*, *Zfp637*, *Zfp710*, *Zfp821*), Kruppel-like factors (*Klf2*, *Klf4*, *Klf9*, *Klf11*), and members of the MYC/MAX/MAD network of transcription factors (*Mxd1*, *Mnt*).

Interestingly, the pathways related to 'response to DNA damage' and 'cellular response to stress' were also enriched (5% of the genes with negative correlation to longevity). A closer examination revealed that the enrichment signal was due to a number of genes involved in apoptosis regulation, including the tumor suppressor TP53 (encoded by *Trp53*), BCL-2 associated X protein BAX (encoded by *Bax*), transcription factor FOXO3 (encoded by *Foxo3*), as well as mitogen-activated protein (MAP) kinase (encoded by *Mapk1*) (*Figure 4C–E*); they were therefore distinct from those genes directly involved in DNA repair (and found above to have positive correlation with longevity). Several other growth signaling factors, such as transforming growth factor beta (TGF-β) receptor (encoded by *Tgfbr3*) and transcription factor JunD (encoded by *Jund*), were also relatively low in longer lived species (*Figure 4F*). In particular, the transcription factor FOXO3 can be activated by oxidative stress (*Essers et al., 2004*), and the genetic variation within the *FOXO3A* gene was found to be strongly associated with human longevity (*Willcox et al., 2008*).

## Genes enriched in network interaction and housekeeping functions

To understand the regulatory network among the top hits, we visualized the protein-protein interaction using the STRING database (*Jensen et al., 2009*). The results revealed significant network interaction among the genes with positive correlation and those with negative correlation (p value $< 10^{-10}$ in both cases; *Figure 3—figure supplement 1*), suggesting that the longevity-correlating genes were indeed functionally related. Next, we analyzed the system level functions of the top hits to determine if they belonged to the known categories of 'Aging genes', 'Essential genes', 'Housekeeping genes' or 'Transcription Factor genes' (*Table 1—source data 1I*). Interestingly, close to 40% of the top hits belonged to the 'Housekeeping genes' (Fisher exact test p value = $3.646 \times 10^{-26}$), whereas the other categories were much less significant (*Table 1—source data 1I*). Therefore, the longevity variation across these species was accompanied by the coordinated modulation of a large number of genes with housekeeping functions in a systemic manner.

## Fibroblast resistance to lethal stresses and toxicity

The longevity-associated expression patterns identified above suggested that the longer lived species might be more efficient at handling and repairing cellular damage. It was previously demonstrated that skin-derived fibroblasts from long-lived rodent species were more tolerant of the stress conditions induced by cadmium, hydrogen peroxide, heat, and DNA alkylating agent methyl methanesulfonate (MMS), and were more resistant to the metabolic inhibition by rotenone treatment and in low-glucose medium (*Harper et al., 2007*). To see if the same effects could be observed in our

study, we subjected the primary fibroblasts to six different stress conditions: treatments with cadmium, hydrogen peroxide, MMS, paraquat, and thapsigargin (inhibitor of sarco/endoplasmic reticulum $Ca^{2+}$ ATPase), as well as to low-glucose culture medium. As expected, the results showed positive correlation between ML and the resistance to cadmium and paraquat (*Figure 4G,H*), although the other conditions did not reach the statistical threshold of p value < 0.05.

## Metabolites correlating with longevity traits

For 12 of the rodent species, we also performed metabolic analyses (*Townsend et al., 2013*) (*Figure 1—source data 1D*). After data filtering and normalization, 144 water-soluble metabolites and 82 lipids were reliably detected across the 22 biological samples (*Supplementary file 2*). Principal Component Analysis (*Figure 5A*) and the phylogram based on metabolite levels (*Figure 5B*) both indicated that the metabolic profiles of these species, like the gene expression, segregated according to phylogeny, although the patterns were less clear-cut than those based on the RNAseq dataset. This might be partly due to the much smaller number of metabolites detected compared to the genes (226 metabolites vs. 9389 genes). Nevertheless, the biological and technical replicates clustered together (*Figure 5B*), suggesting that the within-species variation was relatively small.

To identify the metabolites with significant correlation with the longevity traits, we also applied the phylogenetic regression method described above. At the cut-off of p value.robust < 0.01 (~11% FDR) and p value.max < 0.05, 13 metabolites showed significant correlation with AW, 26 metabolites with ML, 20 metabolites with FTM, 16 metabolites with MLres and 19 metabolites with FTMres (*Figure 5C*, *Table 2—source data 1A–F*). Twenty-three of these metabolites were supported by two or more longevity traits. Pathway analysis revealed the enrichment of 'common amino acids' among the top hits with positive correlation, and 'glycerophospholipids' among the top hits with negative correlation (*Table 2—source data 1G–H*). In particular, several showed positive correlation with multiple longevity traits (*Figure 5D*, *Table 2*); so did a number of nucleotides/nucleosides including ADP, GDP, and adenosine. In terms of negative correlation, a number of lysophosphatidylchonline (LPC; e.g. C16:0 LPC, C18:0 LPC, C18:1 LPC) and lysophosphatidylethanolamine (LPE; e.g. C20:4 LPE, C22:6 LPE) showed significant relationship (*Table 2—source data 1*), which were consistent with the previous report of low LPC and LPE in long-lived mammals (*Ma et al., 2015b*). LPC levels were also previously reported to decrease with age but maintained in mice under caloric restriction (*De Guzman et al., 2013*).

## Validation of amino acid patterns in primate and bird fibroblasts

To further examine our observation of the positive correlation between amino acids and the longevity traits, we independently obtained and quantified the amino acid levels in a larger collection of primary fibroblasts from 15 primate species and 33 bird species. All 10 of the amino acids associated with lifespan in rodent fibroblasts (arginine, glutamate, histidine, leucine, lysine, methionine, phenylalanine, proline, tryptophan, tyrosine, and valine) were also found to have a significant positive association with lifespan in bird and primate fibroblasts (*Table 2*; *Figure 5—figure supplement 1*). The associations were particularly strong for amino acids with hydrophobic side chains. When we adjusted for the effects of body mass, the observed relationships weakened significantly (*Table 2*), likely due to the strong correlation between AW and ML. Nevertheless, the same weakening was also evident in the rodent fibroblasts (*Table 2—source data 1*), indicative of the consistency in the trends. Overall, the positive correlation between fibroblast amino acid levels and species longevity was a feature consistent across rodents, primates, and birds, indicating that some of the longevity signatures identified here may be representative of and generalized to other species.

## Discussion

All lines of mammals descended from the same common ancestor over the previous 230 million years and have since undergone remarkable diversification in body size, metabolic rate, fertility, and longevity, with corresponding changes in the gene expression and metabolite landscape (*Fushan et al., 2015*; *Ma et al., 2015b*). As fibroblasts can be obtained without sacrificing animals and can be cultured under standardized conditions, it is of great interest to determine if their gene expression and metabolite patterns represent lifespan variation across mammals. Fibroblasts are also amenable to experimental manipulation. On the other hand, cross-species gene expression

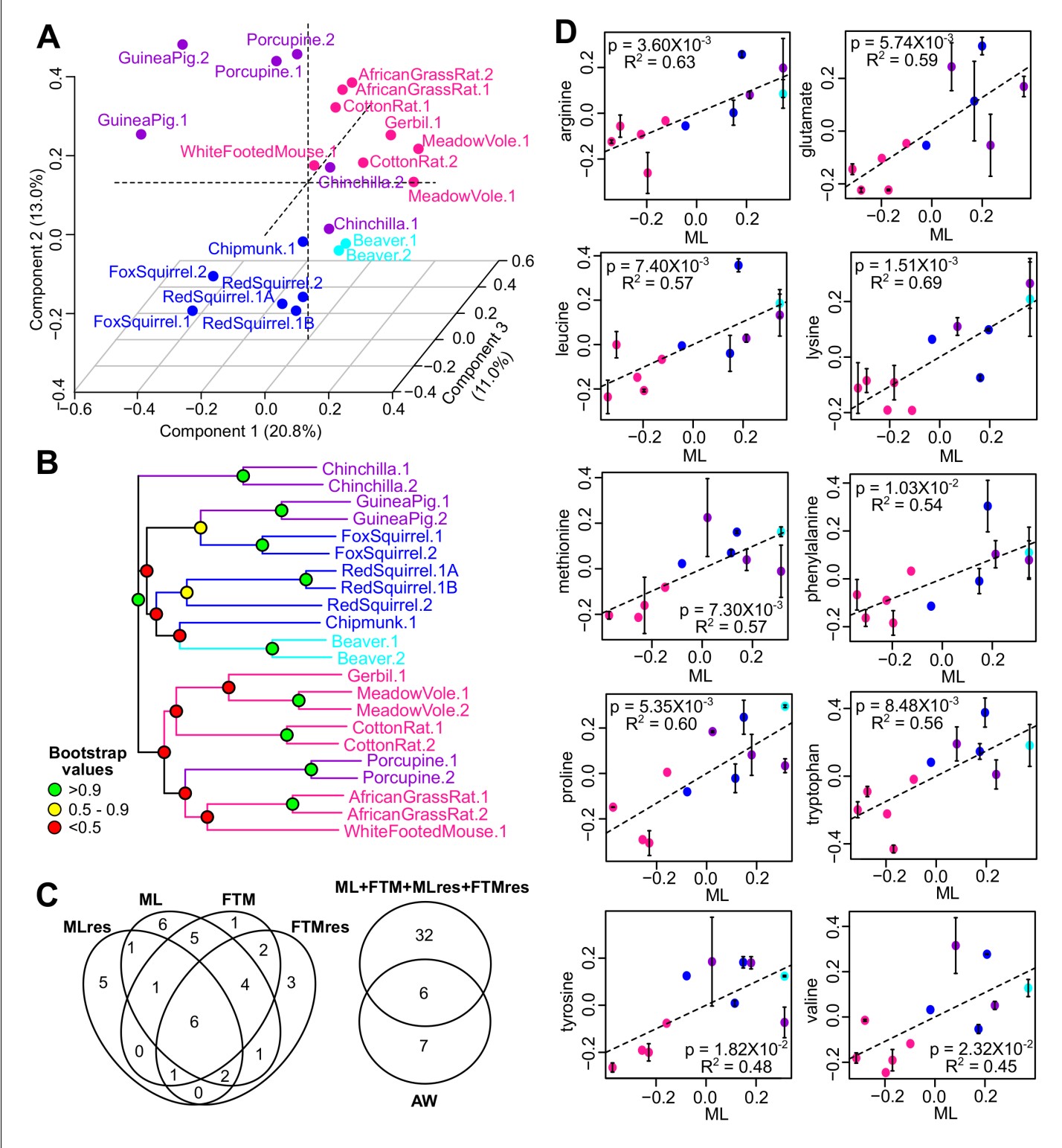

**Figure 5.** Metabolite variation and correlation with longevity. (**A**) Projection of the first three Principal Components (PCs) in Principal Component Analysis. Values in parenthesis indicate percent of variance explained by each of the PCs. Points are colored by taxonomic order (same color scheme as in *Figure 1*) (**B**) Metabolite phylogram. Color of the nodes indicates the result of 1000 times bootstrap. (**C**) Overlap of metabolites associating with Adult Weight and longevity traits. AW: Adult Weight; ML: Maximum Lifespan; FTM: Female Time to Maturity; MLres: Maximum Lifespan Residual; FTMres: Female Time to Maturity Residual. (**D**) Amino acids showing positive correlation with Maximum Lifespan (ML). In each plot, the amino acid levels (vertical axis) and the longevity traits (horizontal axis) are centered at 0 on log10 scale and then transformed by the best-fit variance-covariance

*Figure 5 continued on next page*

*Figure 5 continued*

matrix under phylogenetic regression (i.e. to remove the phylogenetic relationship). The potential outlier point has been removed and the remaining points are shown on the plot and colored by taxonomic group. The regression slope p value (i.e. p value.robust) and $R^2$ value are indicated. Error bars indicate standard error of mean.

The following figure supplement is available for figure 5:

**Figure supplement 1.** Amino acid levels in primate and bird fibroblasts correlate positively with species maximum lifespan.

analyses are often hampered by the lack of publicly available genomes and gene orthology information, especially for those species not commonly studied. Using primary fibroblasts from 16 species of rodents, bats, and shrew, we developed a pipeline for generating species-specific ortholog sets, profiled gene expression by RNAseq and metabolites by mass spectrometry, and identified the molecular features associated with longevity traits.

Our pipeline can be easily extended for a larger number of species. We defined gene orthology based on reciprocal best hit in BLAST (*Tatusov et al., 1997*) and ignored the issues of gene duplication and gene loss. Sequence fragments and missing sequences were filled up using consensus data

**Table 2.** Amino acid levels showing consistent positive correlation with longevity traits.

For the mammalian fibroblast dataset, the number of longevity traits (out of Maximum Lifespan; Female Time to Maturity; Maximum Lifespan Residual; and Female Time to Maturity Residual) with significant positive correlation with the amino acid levels at two different cut-offs (p value.robust < 0.01 and p value.robust < 0.05) are shown. For the primate and bird fibroblast dataset, the regression was performed using primate data only, bird data only, and the pooled data of both. The regression slope p value < 0.05 are in bold.

| | Mammalian fibroblasts | | Primate and bird fibroblasts | | | | | |
| | No. of longevity traits (out of four) with significant correlation | | Regression slope p value with species maximum lifespan | | | Regression slope p value with species maximum lifespan residual | | |
| Amino acid | p value. robust < 0.01 | p value. robust < 0.05 | Primates only | Birds only | Primates and birds | Primates only | Birds only | Primates and birds |
| --- | --- | --- | --- | --- | --- | --- | --- | --- |
| arginine | 3 | 4 | **$3.4 \times 10^{-2}$** | $8.6 \times 10^{-2}$ | **$3.1 \times 10^{-2}$** | $3.8 \times 10^{-1}$ | **$1.1 \times 10^{-2}$** | **$2.1 \times 10^{-2}$** |
| glutamate | 2 | 4 | $6.5 \times 10^{-2}$ | **$1.8 \times 10^{-2}$** | **$1.1 \times 10^{-2}$** | **$4.6 \times 10^{-2}$** | $2.8 \times 10^{-1}$ | $1.3 \times 10^{-1}$ |
| histidine | 0 | 4 | $9.4 \times 10^{-2}$ | $6.0 \times 10^{-2}$ | **$4.3 \times 10^{-2}$** | $2.3 \times 10^{-1}$ | $1.4 \times 10^{-1}$ | $1.7 \times 10^{-1}$ |
| leucine | 2 | 4 | **$2.9 \times 10^{-3}$** | $6.0 \times 10^{-2}$ | **$4.8 \times 10^{-3}$** | **$1.4 \times 10^{-2}$** | $5.9 \times 10^{-1}$ | $2.3 \times 10^{-1}$ |
| lysine | 3 | 3 | **$9.8 \times 10^{-3}$** | $8.2 \times 10^{-2}$ | **$1.4 \times 10^{-2}$** | $9.1 \times 10^{-2}$ | $2.9 \times 10^{-1}$ | $2.5 \times 10^{-1}$ |
| methionine | 1 | 3 | $3.2 \times 10^{-1}$ | **$1.4 \times 10^{-2}$** | **$2.7 \times 10^{-2}$** | $3.0 \times 10^{-1}$ | **$3.0 \times 10^{-2}$** | **$4.9 \times 10^{-2}$** |
| phenylalanine | 1 | 4 | **$9.8 \times 10^{-3}$** | **$1.2 \times 10^{-3}$** | **$2.1 \times 10^{-4}$** | $8.2 \times 10^{-2}$ | $1.3 \times 10^{-1}$ | $1.2 \times 10^{-1}$ |
| proline | 1 | 4 | **$4.4 \times 10^{-3}$** | **$3.9 \times 10^{-4}$** | **$3.6 \times 10^{-5}$** | **$3.5 \times 10^{-2}$** | $1.2 \times 10^{-1}$ | **$5.4 \times 10^{-2}$** |
| tryptophan | 2 | 4 | **$9.2 \times 10^{-3}$** | **$7.8 \times 10^{-4}$** | **$1.2 \times 10^{-4}$** | **$2.6 \times 10^{-2}$** | $2.5 \times 10^{-1}$ | $1.5 \times 10^{-1}$ |
| tyrosine | 1 | 3 | $3.2 \times 10^{-1}$ | **$8.8 \times 10^{-3}$** | **$1.8 \times 10^{-2}$** | $4.3 \times 10^{-1}$ | $1.7 \times 10^{-1}$ | $2.9 \times 10^{-1}$ |
| valine | 0 | 3 | **$1.2 \times 10^{-2}$** | **$5.4 \times 10^{-3}$** | **$1.0 \times 10^{-3}$** | $2.0 \times 10^{-1}$ | $2.8 \times 10^{-1}$ | $3.2 \times 10^{-1}$ |

**Source data 1.** Phylogenetic regression of metabolite levels against longevity traits. Regression against (A) Adult Weight; (B) Maximum Lifespan; (C) Female Time to Maturity; (D) Maximum Lifespan Residual; and (E) Female Time to Maturity Residual. 'coef.all', 'p value.all', and 'q value.all' refer to the regression slope, p value, and FDR-adjusted q value using all the species. 'p value.robust' and 'q value.robust' refer to the statistics after removing the potential outlier point. 'p value.max' and 'q value.max' refer to the maximal (least significant) regression p value and q value when each one of the species was left out, one at a time. Only genes with p value.robust < 0.01 and p value.max < 0.05 are shown. (F) Top hits identified by two or more longevity traits. The p value.robust against each of the four longevity traits (ML, FTM, MLres, and FTMres) as well as adult weight (AW) are shown. These metabolites were the input for pathway enrichment analysis. Pathway enrichment analysis of metabolites showing (G) positive and (H) negative correlation with longevity traits. Enrichment was performed based on hypergeometric statistics. (I) Top hits identified by two or more longevity traits, using cut-off of p value.robust < 0.05. The p value.robust against each of the four longevity traits (ML, FTM, MLres, and FTMres) as well as adult weight (AW) are shown.

from the other species. While these steps unavoidably introduced inaccuracy within our species-specific ortholog sequences, they did not affect the overall read alignment results (*Figure 2B–C*, *Figure 2—figure supplement 1*). Furthermore, we observed no significant differences in sequence divergence between those ortholog sets showing correlation to longevity and those that did not (Wilcoxon Rank Sum Test p value = 0.32; *Figure 2—figure supplement 1D*), so the degree of longevity correlation was not biased by the degree of sequence conservation.

The gene expression findings revealed a clear segregation based on phylogeny (*Figure 3A–B*), suggesting that evolutionary relationships significantly influenced the expression patterns. On the other hand, the metabolite patterns were less clear-cut (*Figure 5A–B*), which might be attributed to the smaller number of species and metabolites. Using phylogenetic regression and a two-step verification procedure, we identified a list of genes and metabolites with significant correlations to multiple longevity traits. The pathways of 'nucleotide binding', 'DNA repair', 'chromosome organization', and 'glucose metabolic process' were enriched among the genes with positive correlation with longevity, whereas 'proteolysis', 'protein transportation/localization' and 'regulation of transcription' were enriched for genes showing a negative correlation. Furthermore, a significant number of these longevity-correlating genes are involved in 'housekeeping' functions, implying that lifespan variation across species is often accompanied by coordinated shifts in the gene expression landscape and modulation of fundamental biological processes.

The link between proteolysis/autophagy and aging has been proposed by a number of authors. In general, proteolytic functions decline and oxidized proteins increase with age, and autophagy genes are required for the lifespan extension effects of Insulin/IGF-1 signaling and dietary restriction (*Chondrogianni and Gonos, 2008*; *Hansen et al., 2008*; *Kenyon, 2010*; *Kevei and Hoppe, 2014*; *Löw, 2011*; *Meléndez et al., 2003*; *Rubinsztein et al., 2011*; *Starke-Reed and Oliver, 1989*; *Vernace et al., 2007*). Activation of proteasome or autophagy has been shown to extend lifespan in *C. elegans* (*Chondrogianni et al., 2015*; *Ghazi et al., 2007*), yeast (*Kruegel et al., 2011*), and flies (*Simonsen et al., 2008*). Immunoproteasome and proteasome activity was also elevated in the livers of long-lived Snell dwarf mice and in mice exposed to drugs known to extend lifespan (*Pickering et al., 2015*). On the other hand, our results suggest that fibroblasts from the longer lived animals actually have lower expression levels of genes involved in proteolysis, autophagy, and apoptosis but higher expression of genes related to DNA repair and maintenance. In particular, the genes coding for the tumor suppressor TP53, apoptosis regulator BAX, and several growth and proliferation signaling pathways were all down-regulated in the fibroblasts of the longer lived species (*Figure 4C–F*). One possible interpretation may be that the longer lived species generate less damage and/or have better repair mechanisms, so that the cells rely less on proteolysis, autophagy and apoptosis. Previous studies reported enhanced DNA repair capacity and reduced oxidative damage in longer lived species (*Adelman et al., 1988*; *Cortopassi and Wang, 1996*; *Grube and Bürkle, 1992*; *Perez-Campo et al., 1998*; *Sohal et al., 1995*). Down-regulation of the ubiquitin ligase complex was also reported in the liver of longer lived mammalian species (*Fushan et al., 2015*). In agreement, our toxicology experiments confirmed that the fibroblasts of longer lived species were more resistant to oxidative stress induced by cadmium and paraquat treatments. In terms of metabolites, the pattern of low LPC and LPE among long-lived species was consistent with previous reports, and the positive correlation between amino acids and longevity was independently validated using fibroblasts from multiple species of primates and birds, suggesting that these changes in cell biology are likely to have evolved, independently, in each of these separate lineages in association with slower aging and longer lifespan.

On the other hand, several possible caveats in our data warrant additional attention. The levels of gene expression could be influenced by confounding factors such as gene length and proximity to other genes (*Chiaromonte et al., 2003*), and our definition of orthology might have missed out those genes with less conserved sequences. Our analyses were limited to those genes expressed in fibroblasts, and the effects of different spliced isoforms were not captured by our data. Furthermore, although our RNA sequencing and metabolic measurements were performed on cells of second or third passage, it was nevertheless possible that in vitro culture conditions might have introduced changes in chromatin architecture (*Zhu et al., 2013*), or that differences in stress-related pathways might reflect differential responses to the culture conditions. In addition, the use of ML as an aging research metric can be problematic for the less well documented animals due to the small sample size, high variance, and reliance on a single individual per species (*Moorad et al., 2012*). Although

FTM is less prone to reporting bias and shows strong correlation with ML (Spearman correlation coefficient 0.87), it may be influenced by seasonality factors. Other parameters with better statistical properties, such as mean adult lifespan and 90th quantile of longevity (*Moorad et al., 2012*), should be used in place of ML in the long run, although at the moment such records are available for only a limited number of species.

Overall, our study supports the idea that gene expression, and to some degree metabolite levels, in fibroblast cultures can uncover differences in cell biology and metabolism that correspond to longer life. Apparently, these expression patterns are preserved when the intraorganismal environment is removed and cells instead are subjected to standardized cell culture conditions in the lab setting. This makes fibroblasts a particularly attractive experimental system to examine and manipulate molecular patterns, with gene expression (or metabolite patterns) as a readout. While our study represents an initial study, this approach can be extended to a larger group of species and samples, refining the molecular signatures and then manipulating them via genetic and environmental manipulations. Ultimately, this should reveal the genetic basis for differences in species longevity and lead to new strategies for targeting them, thereby shifting cells, and ultimately organisms, to the state of cells from related longer-lived species.

# Materials and methods

## Sample collection

Primary skin fibroblast samples were collected from shrew (*Blarina brevicauda*), big brown bat (*Eptesicus fuscus*), little brown bat (*Myotis lucifugus*), guinea pig (*Cavia porcellus*), porcupine (*Erethizon dorsatum*), chinchilla (*Chinchilla lanigera*), chipmunk (*Tamias striatus*), fox squirrel (*Sciurus niger*), red squirrel (*Sciurus vulgaris*), beaver (*Castor canadensis*), gerbil (*Meriones unguiculatus*), African grass rat (*Arvicanthis niloticus*), meadow vole (*Microtus pennsylvanicus*), cotton rat (*Sigmodon hispidus*), white-footed mouse (*Peromyscus leucopus*), and deer mouse (*Peromyscus maniculatus brandii*) (*Figure 1—source data 1*). The post-pubertal animals (gender and ages were not recorded) were caught opportunistically in an area extending approximately 400 km north and 80 km south of Ann Arbor, MI, USA (*Harper et al., 2007*). Abdominal skin areas were sterilized with 70% ethanol wipes and biopsies of at least 5 mm by 5 mm in area were obtained and placed in complete media (CM) made of Dulbecco's modified Eagle medium (DMEM, high-glucose variant, Gibco-Invitrogen, Carlsbad, CA) supplemented with 20% heat-inactivated fetal bovine serum, antibiotics (100 U mL$^{-1}$ penicillin and 100 μg mL$^{-1}$ streptomycin; Sigma, St. Louis, MO) and 0.25 μg mL$^{-1}$ of fungizone (Biowhittaker-Cambrex Life Sciences, Walkersville, MD) on ice and shipped overnight to our laboratory (*Harper et al., 2007*). Biological replicates (i.e. tissues from different individuals) and technical replicates were collected on selected species (*Figure 1—source data 1*).

## Cell culture

The conditions for establishment and maintenance of the cultures have been reported previously (*Harper et al., 2007*; *Murakami et al., 2003*; *Salmon et al., 2005*). Briefly, trypsinized cells were grown to 90% confluence, and we found no significant differences among species in the interval between initial seeding and initial confluence (*Harper et al., 2007*). Cells were then harvested and placed in a new culture flask, fed at days 3 and 7, and then subcultured to a fixed density of $7.5 \times 10^5$ cells for 75 cm$^2$ flask. These cells were then harvested 7 days later and cryopreserved at $10^6$ cells per vial.

Production of cells for RNA sequencing and metabolite profiling always started by thawing a vial of cryopreserved cells and allowing them to expand until the culture had produced sufficient cells (at least $30 \times 10^6$ cells) for analysis. These cultures were kept under low-oxygen conditions (3% O$_2$) after thawing to minimize selection for resistance to O$_2$ toxicity (*Busuttil et al., 2003*; *Parrinello et al., 2003*). Cells were harvested using trypsin and pelleted by centrifuging for 5 min at 230 rcf. After counting, the cells were divided into aliquots of $10 \times 10^6$ cells. Two washes with PBS (-Ca,-Mg) were performed, any excess PBS was drained, and the pellets were frozen at −80°C. Technical replicates were made by growing a minimum of $60 \times 10^6$ cells and labeling half of the cells after counting as separate samples.

## Life history data of the species

The Adult Weight (AW), Maximum Lifespan (ML) and Female Time to Maturity (FTM) data of the species (or if not available, for a closely related species) were obtained from the Animal Ageing and Longevity (AnAge, RRID:SCR_001470) Database (*Tacutu et al., 2013*). In addition, since both ML and FTM increase with AW, we calculated the body mass adjusted residuals (i.e. MLres and FTMres), to represent the ratio between the observed longevity and the expected longevity based on body mass (*Ma et al., 2015b*; *Tacutu et al., 2013*). Two allometric equations were used to calculate the residuals. The MLres equation, $MLres = ML/(4.88 \times AW^{0.153})$, was based directly on the documentation of the AnAge database (http://genomics.senescence.info/help.html#anage). The FTMres equation, $FTMres = FTM/(78.1 \times AW^{0.217})$, was based on linear regression using the FTM and body mass records of 1330 mammalian species in the AnAge database.

## RNA sequencing

RNAseq libraries were prepared as previously described (*Fushan et al., 2015*). Paired end sequencing was done on the Illumina HiSeq2000 platform generating approximately 30 to 75 million reads per sample, with read length 50 or 100 nucleotides (*Figure 1—source data 1*). The raw data were processed by Cutadapt (RRID:SCR_011841) (*Martin, 2011*) to remove low-quality reads.

## Species-specific ortholog sets and expression values

Reference genomes were publicly available for five species (*Eptesicus fuscus, Myotis lucifugus, Cavia porcellus, Chinchilla lanigera, Peromyscus maniculatus brandii*). To ensure consistency across the entire dataset, we developed the following pipeline to identify species-specific ortholog sets, map the reads and obtain expression values (*Figure 2*).

Step 1: generate mouse reference. Based on the *Mus musculus* Ensembl genome and annotation (release 78) (RRID:SCR_006773), the longest transcript was extracted for each protein-coding gene locus, after confirming the presence of start and stop codons and the proper reading frame. Those transcripts containing highly repetitive or highly similar sequences were identified and removed using BLAST (RRID:SCR_004870) (at e-value cut-off $10^{-6}$) (*Camacho et al., 2009*). This generated the Mouse Reference, representing the coding sequences of 16,816 unique protein-coding genes (*Supplementary file 1*).

Step 2: identify species-specific ortholog sets. For each species, the transcriptome was assembled de novo from the RNAseq reads using Trinity (RRID:SCR_013048) (*Grabherr et al., 2011*). BLAST (with 'dc-megablast' option) was performed between Mouse Reference and the assembled transcriptome (and the published genome, if available) of each species to identify the reciprocal best hits (*Tatusov et al., 1997*). The sequences were trimmed down to open reading frame (i.e. flanked by start and stop codons) using Exonerate (*Slater and Birney, 2005*). Within each ortholog sets, multiple sequence alignment was performed using MUSCLE (RRID:SCR_011812) (*Edgar, 2004*) and the percentage of sequence identity was assessed by MVIew (*Brown et al., 1998*). For the sequence fragments or missing sequences due to poor coverage, they were filled up using the consensus. We confirmed the filling up procedure did not significantly affect the read counting results (*Figure 2—figure supplement 1*). Seventy-four percent of the ortholog sets did not require filling up or were filled up <10% of the sequence length, whereas 5% of the ortholog sets were filled up 90–100% of the sequence length (*Figure 2—figure supplement 1A*). When the expression values were standardized to mean = 0 and standard deviation = 1 within each ortholog set, there was no significant bias against those ortholog sets with high percentage of filling up (*Figure 2—figure supplement 1B*).

Step 3: read mapping, counting, filtering and normalization. The RNAseq reads were mapped to the species-specific ortholog sets using STAR (*Dobin et al., 2013*), with an average read alignment rate of ~40% (*Figure 1—source data 1*). As comparison, read mapping to publically available genomes achieved an average alignment rate of ~85% (*Figure 1—source data 1*). The lower alignment rate to the species-specific ortholog sets was likely due to the exclusion of 5' and 3' untranslated regions, repetitive or highly similar sequences, and introns. Nevertheless, the alignment rates were largely similar across the samples and species (*Figure 2C*). Read counting was performed by featureCounts (RRID:SCR_012919) (*Liao et al., 2014*) and those ortholog sets with too high counts (i.e. read counts contributing to >5% of the total counts; three orthologs were removed this way) or too low counts (i.e. <10 counts in four or more samples) were discarded. The library sizes were then

scaled by trimmed mean of M-values (TMM) method, log10-transformed, and quantile-normalized (*Robinson and Oshlack, 2010*). The final expression set consisted of 9389 gene orthologs across 28 samples. Shapiro Test confirmed normalcy assumption was valid for 89% of the genes on log10 scale. The pairwise DNA distance within each ortholog set was calculated based on the Kimura 2-parameters distance (*Kimura, 1980*).

### Metabolite profiling and data processing

For rodent cells, the metabolite levels were quantified by mass spectrometry as previously described (*Townsend et al., 2013*). From the raw metabolite measurements, we only kept metabolites with <10% missing values. The raw values were normalized separately for the three collection modes (water soluble positive ionization mode 'HILIC-pos', water soluble negative ionization mode 'HILIC-neg', and lipid mode 'C8-pos'), first by the internal standards, and then by the total signals within each mode. The data were then log10-transformed and quantile normalized. The final expression set consisted of 226 metabolites across 22 samples. Shapiro Test confirmed normalcy assumption was valid for 90% of the metabolites on log10 scale.

### Principal component analysis and phylograms

Principal component analysis was performed on the standardized expression values or metabolite values and the first three Principal Components were extracted. The phylograms were constructed using the neighbor joining method (*Saitou and Nei, 1987*), based on the distance matrix of 1 minus Pearson correlation coefficient of the standardized expression values or metabolite values. The reliability of the branching patterns was assessed by 1000 times bootstrap.

### Phylogenetic regression and two-step verification procedure

To identify genes or metabolites with significant correlation to the longevity traits, regression was performed using the generalized least square approach, by incorporating the phylogenetic relationship in the variance-covariance matrix (*Felsenstein, 1985*; *Ma et al., 2015b*; *Revell, 2012*). As previously described (*Ma et al., 2015b*), four different trait evolution models ('null', 'Brownian motion', 'Pagel's lambda', and 'Ornstein-Uhlenbeck') were tested and the best fit model was selected based on maximum likelihood.

A two-step procedure was applied to verify the robustness of the results (*Ma et al., 2015b*). In the first step, the species whose exclusion would lead to most improvement in the slope p value (i.e. a potential outlier), was identified and removed. The regression p value of this step was reported as 'p value.robust'. In the second step, each of the remaining species was removed, one at a time, and the regression was repeated. The largest (i.e. least significant) p value of this step was reported as 'p value.max'. The False Discovery Rate adjusted values were 'q value.robust' and 'q value.max', respectively. To qualify as a top hit, we required a gene to have p value.robust < 0.01 and p value.max < 0.05. For pathway enrichment purposes, we further required that the genes were identified as a top hit in two or more longevity traits (ML, FTM, MLres or FTMres).

### Pathway enrichment analysis and interaction network

For the genes, pathway enrichment analysis was performed using DAVID (RRID:SCR_003033) (*Huang da et al., 2009a*, *2009b*). The 16,816 unique protein-coding genes in Mouse Reference were set as the background and the pathways were based on *Mus musculus*. For those genes showing positive and negative correlation with longevity (supported by two or more longevity traits), we queried Gene Ontology ('GO Term'; Biological Process and Molecular Functions only), SwissProt and Protein Information Resource ('SP PIR Keywords'), and Kyoto Encyclopedia of Genes and Genomes ('KEGG Pathway'). For comparison, pathway enrichment was also performed using only the 9389 expressed orthologs as background. STRING (RRID:SCR_005223) version 10 (*Jensen et al., 2009*) was used to visualize the interaction network among the top hits, based on the mouse genes. The required score was set as 400 and the network output was generated using R package (RRID: SCR_006442) 'STRINGdb'. Selected nodes were highlighted based on the enriched pathways.

We also analyzed the association between longevity-associated genes (either positively or negatively or both) and each of four functional groups of genes – aging genes, essential genes, transcription factor genes, and housekeeping genes. These human gene sets were originally collected and

analyzed in a previous study (*Zhang et al., 2016*). Human aging genes were obtained from GenAge (RRID:SCR_010223) (build 17) (*de Magalhães and Toussaint, 2004*). They include both genes related to fundamental human aging processes and those associated with human longevity based on evidence from human and model organisms. Human essential genes are the human orthologs of mouse essential genes, whose deletions result in prenatal, prenatal or postnatal lethality in mouse. Human transcript factor genes were from Panther database (RRID:SCR_004869) (*Mi et al., 2013*). Human housekeeping genes were from (*Eisenberg and Levanon, 2013*). Housekeeping genes are considered to be involved in basic cell maintenance, and thus expected to maintain relatively constant expression levels in different cells and conditions (*Eisenberg and Levanon, 2013*). For the enrichment analysis, we used two different sets of background genes. One includes all the orthologs tested (i.e. 16,816 genes), and the other only genes expressed in fibroblasts (i.e. 9389 genes). All mouse genes were mapped to their human orthologs through Ensembl BioMart (RRID:SCR_002344) (*Smedley et al., 2015*), and only genes with one to one mapping relationship were used. In this analysis, we used 14,749 and 8809 human orthologs as background genes. Enrichment statistics were based on Fisher's exact test.

For the metabolites, pathway information was obtained from ConsensusPathDB (RRID:SCR_002231) (*Kamburov et al., 2009*) and Human Metabolome Database (HMDB) (RRID:SCR_007712) (*Wishart et al., 2013*). For ConsensusPathDB, only pathways with known KEGG IDs were incorporated. Analysis was performed on pathways with at least 5 but less than 100 metabolites. Enrichment statistics was based on a hypergeometric distribution (*Tavazoie et al., 1999*). Odd ratios and expected counts were calculated as previously described (*Gentleman et al., 2013*).

## Evaluation of amino acid levels in bird and primate fibroblast cell lines

An untargeted metabolomics screen was conducted using fibroblasts from 32 bird species and 13 species of non-human primates. The detailed methods were described in *McDonnell et al. (2013)*. Briefly, frozen cell cultures were homogenized in water, volumes were adjusted based on protein concentration in the extract, and proteins were precipitated by ethanol containing recovery standards. Each extract was split into two aliquots (for LC-MS and GC-MS), dried down. LC-MS aliquot was re-suspended in water containing injection standards and analyzed on an Agilent 1200 LC/6530 qTOF LC-MS system using a Waters Acquity HSS T3 C18 column. GC-MS aliquot was derivatized by BSTFA and analyzed on Agilent 7890 A-5975C inert XL MSD GCMS instrument using HP-5MS 5% phenyl-methyl siloxane column (30m x 250 μm x 0.25 μm). Data extraction and analysis was performed using Agilent MassHunter Qualitative Analysis software and in-house metabolite libraries.

Although the original dataset contained information on 4383 metabolites, including 456 of known chemical identity, the analysis for this paper was restricted to the ten amino acids for which $p < 0.05$ for association with maximum lifespan in the analysis of mammalian (i.e. rodent, shrew and bat) fibroblasts (*Figure 5* and *Table 2—source data 1*). A regression analysis was performed using a model in which the dependent variable was the logarithm of the species maximal lifespan, and the independent variables were the metabolite level and a categorical variable reflecting whether the species was bird or primate. The procedure tested the association between metabolite and lifespan in the entire set of species, but did not make the assumption that the slope was necessarily the same for birds as for primates. The resulting F-statistic was evaluated for significance based upon an empirically generated set of null distributions (for each metabolite) by permutation. When two or more of the untargeted features were annotated as corresponding to the same amino acid (seven cases), we tabulated the degree of association from the feature most strongly correlated with lifespan among the species studied. Although these would introduce some bias in favor of correlation, we noted that in four out of the seven cases, the multiple features were all significant at p value < 0.05; in the other three cases, the features were all significant at p value < 0.2. Parallel analyses were also done for bird species and for non-human primate species independently.

## Resistance of rodent fibroblasts to lethal and metabolic stresses

The methods used are as previously described (*Harper et al., 2007*; *Murakami et al., 2003*; *Salmon et al., 2005*). Briefly, each test procedure began by culturing the cells at a density of $3 \times 10^4$ cells in 100 μL CM in 96-well microtiter plates for 24 hr, followed by a period of 24 hr in medium lacking serum but containing 2% bovine serum albumin (BSA, Sigma) with antibiotics and

fungizone at the same concentration as CM. For assessment of resistance to $H_2O_2$, paraquat, and cadmium (Sigma), the cells in the 96-well plates were washed and exposed to the stress agent for 6 hr. For assessment of resistance to methyl methanesulfonate (MMS), the cells were incubated with MMS in DMEM for 24 hr, washed and then incubated with DMEM supplemented with 2% BSA, antibiotics, and fungizone for 18 hr. For assessment of cell metabolism in low-glucose medium, cells were incubated in DMEM containing a range of glucose concentrations for 1 hr. Survival was assessed by WST-1 tests. All incubations were at 37°C in a humidified incubator with 5% $CO_2$ in air.

For calculation of the resistance of each cell line to chemical stressors, at each dose of chemical stressor, mean survival was calculated for triplicate wells for each cell line. The LD50, i.e. dose of stress agent that led to survival of 50% of the cells, was then calculated using Probit analysis as implemented in NCSS software (NCSS, Kaysville, UT). ED50 values for glucose withdrawal were calculated in a similar manner to estimate the level of glucose or rotenone associated with a 50% reduction in cellular metabolic activity.

## Acknowledgements

We wish to thank William Kohler and Melissa Han for development of cell lines and preparation of cell pellets. Supported by NIH AG047745, AG023122, AG047200, DK089503, DK097153, and Life Extension Foundation.

## Additional information

### Funding

| Funder | Grant reference number | Author |
|---|---|---|
| National Institutes of Health | DK097153 | Charles F Burant |
| National Institutes of Health | AG047200 | Zhengdong D Zhang<br>Andrei Seluanov<br>Vera Gorbunova<br>Vadim N Gladyshev |
| National Institutes of Health | AG047745 | Vadim N Gladyshev |
| National Institutes of Health | AG023122 | Vadim N Gladyshev |
| National Institutes of Health | DK089503 | Vadim N Gladyshev |
| Life Extension Foundation | | Vera Gorbunova<br>Andrei Seluanov |

The funders had no role in study design, data collection and interpretation, or the decision to submit the work for publication.

### Author contributions

SM, RAM, Conception and design, Acquisition of data, Analysis and interpretation of data, Drafting or revising the article; AU, AG, Y-MT, CFB, SR, QZ, ZDZ, AS, VG, CBC, Acquisition of data, Analysis and interpretation of data, Drafting or revising the article; VNG, Conception and design, Analysis and interpretation of data, Drafting or revising the article

### Author ORCIDs

Vadim N Gladyshev, http://orcid.org/0000-0002-0372-7016

## Additional files

### Supplementary files

• Supplementary file 1. Gene expression values. (A) Raw counts. (B) log10 normalized values.

• Supplementary file 2. Metabolite levels. (A) Raw values. (B) log10 normalized values.

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
