## [Decision Letter]

Thank you for submitting your article "Cell culture-based profiling across mammals reveals DNA repair and metabolism as determinants of species longevity" for consideration by *eLife*. Your article has been favorably evaluated by Janet Rossant as the Senior Editor and three reviewers, one of whom is a member of our Board of Reviewing Editors. The following individuals involved in review of your submission have agreed to reveal their identity: Daniel Promislow (Reviewer #2) and Yousin Suh (Reviewer #3).

The reviewers have discussed the reviews with one another and the Reviewing Editor has drafted this decision to help you prepare a revised submission.

Summary:

The manuscript by Gladyshev and colleagues seeks to correlate gene expression and metabolite signatures in cultured fibroblasts from 16 different mammals with longevity phenotypes of the respective mammals, including maximal lifespan, body size, female time to maturity, and stress resistance. Using RNAseq and mass spec, the authors show that the constructed gene expression profiles reflect phylogenetic relationships, and identify subsets of genes and metabolites that correlate, positively or negatively, with multiple longevity traits (but not necessarily body size). The study presents a massive amount of work, particularly in compiling the various datasets and developing the informatics pipelines to specify species-specific ortholog sets and assess the robustness of the results, which will certainly be of interest to the community of researchers studying the biology of aging. All three reviewers agreed that the study provides interesting insights into aging and longevity, and should be of broad interest. However, as detailed below, a number of concerns were also raised, relating in large part to insufficient clarity in the current version with respect to the authors' methods and the limitations of their approach and data.

Essential revisions:

1) As this fibroblast culture system is the cornerstone of the authors' approach, it is essential that they provide clear and complete details regarding the experimental strategies used to isolate and culture these cells. In particular, detailed answers to the following key questions must be available in the present manuscript:

A) What age and sex were the animals from which the fibroblasts were isolated (e.g., was% max lifespan of donor animals matched for the different species? Were they sex-matched?)?

B) From what region of the body were fibroblasts isolated? This is important since other studies have shown that fibroblasts retain a regional gene expression program (e.g., hox patterning) after isolation.

C) What is the media system used, and strategy for cell passaging? How were these culture conditions chosen, and how can it be determined if they are optimal for all species (e.g., the authors recently published on a specific sensitivity to oxygen tension for fibroblasts from lab mice, but not other rodent species). This information should be very clearly articulated and discussed in the manuscript text itself, since it forms the basis for the experimental system used.

D) At what passage number were gene expression and metabolite profiles generated?

E) What is the in vitro proliferation rate of the different fibroblast isolates, and does this correlate with any of the gene expression profiles or longevity traits analyzed?

2) It should also be discussed that in vitro cell culture conditions dramatically change chromatin architecture (Zhu et al., Cell 2013), and so gene expression patterns. The gene expression signature detected may simply reflect the ability of cells to differentially adjust to the in vitro conditions, and may contribute to the multiplex stress resistance. This caveat should be discussed in the text.

3) It is not clear whether the common 9,389 gene orthologs that were reliably detected across the 15 species are comparable to the number of expressed genes detected by RNA-seq in fibroblasts in the 5 species with annotated genomes. The base line information on how many transcripts are detected in the 5 species should be available as they may provide a way to estimate the number of potentially missing orthologs, due to sequence divergence, some of which may be critical contributors to longevity through drastic functional alterations of gene products. The text should be clarified to address this point.

4) It is not clear what the baseline gene set was for the enrichment analysis of the 827 top hits (Figure 3). Was it 9,389 genes or whole gene sets? And if the latter, which species? The data should be presented to show the enriched pathways in the top hits using the whole gene sets vs. 9,389 genes as baseline. This is because if DNA repair pathways are already enriched among the 9,389 genes, the conclusion that it is a longevity-associated molecular signature can be misleading. It is formerly possible that genes in these ancient pathways (such as DNA repair or glucose metabolism) may be more conserved sequence-wise and therefore more enriched among the orthologs, and thus among top hits. The text and data presentation should be clarified to address this point.

5) Subsection “Longevity trait variation across mammal”. The authors need to provide more information on how they calculated residuals from body mass. The reference to Ma et al. 2015b gives allometric equations, but no information is provided regarding where those equations come from. The description in Ma et al. (residual LS = LS/(a x Mass^b^)) implies that residuals were taken from a least-squares regression of log(LS) vs. log(Mass) with an intercept of log(a) and a slope of b. But this is not mentioned in the present manuscript, and must be inferred from the Ma et al. manuscript. If a least-squares regression approach was used, one important assumption is that the residuals are normally distributed. The data for Figure 1 (MLres and FTMres) do not look normally distributed. If they are not, a general linear model with an appropriate link function must be applied.

6) The authors use maximum lifespan as a metric of aging, and thus should add to the text of their manuscript a discussion of the concerns that have been raised regarding this statistic (e.g. Moorad et al. 2012 Aging Cell). In particular, it does not provide a measure of aging per se, is quite sensitive to small sample size, and is not in itself under direct selection.

7) Analyses of primate and bird species (subsection “Validation of amino acid patterns in primate and bird fibroblasts”) does not appear to be size corrected. Numerous studies have shown strong size effects of life span not only in primates, but also in birds. Therefore, size effects should be addressed here as well.

8) Subsection “Evaluation of amino acid levels in bird and primate fibroblast cell lines”. For the bird/primate analysis, the authors state, "when two or more […] features were annotated as corresponding to the same amino acid, we tabulated the degree of association from the feature most strongly correlated with lifespan among the species studied." This approach will necessarily bias the result in favor of suggesting a stronger relationship between lifespan and metabolite levels than might be true, and this caveat must be mentioned.

---

## [Author Response]

*Essential revisions:*

*1) As this fibroblast culture system is the cornerstone of the authors' approach, it is essential that they provide clear and complete details regarding the experimental strategies used to isolate and culture these cells. In particular, detailed answers to the following key questions must be available in the present manuscript:*

We agree that these are very important points. We have updated the Methods section to incorporate these reviewers’ comments and suggestions. Additional comments are provided below.

*A) What age and sex were the animals from which the fibroblasts were isolated (e.g., was% max lifespan of donor animals matched for the different species? Were they sex-matched?)?*

Fibroblasts were isolated from post-pubertal adults. Although ages of the wild-captured donors could not be reliably determined, it is a good assumption that most were young, i.e. their age was substantially less than the maximal lifespan that could be achieved by protected or captured members of the species. For mice, for example, maximum lifespan in the laboratory is approximately 3.5 to 4 years, but mice in their natural environment typically live about six months. Thus, most wild-captured adults were likely to be well below half of the species-specific maximal recorded longevity. The sex of the donors was not recorded. Variations in donor age or sex are unlikely to have generated false positive findings in our design. If such variations (age or sex, for example) did have any effects on our endpoints, such effects would have increased Type II error (false negatives), but not have led us to false positive claims (Type I error), unless these demographic variables were confounded with species life history variables. For example, if most of the animals captured from short-lived species were males, and most of those captured from longer-lived species were females, this could in principle create a false positive error by confounding species lifespan with gender effects, but such a scenario seems unlikely.

*B) From what region of the body were fibroblasts isolated? This is important since other studies have shown that fibroblasts retain a regional gene expression program (e.g., hox patterning) after isolation.*

We used an identical procedure to prepare cells. Fibroblasts were taken from sun-protected abdominal skin, as previously reported (Harper et al., Aging Cell 2007). They were therefore consistent across all our species.

*C) What is the media system used, and strategy for cell passaging? How were these culture conditions chosen, and how can it be determined if they are optimal for all species (e.g., the authors recently published on a specific sensitivity to oxygen tension for fibroblasts from lab mice, but not other rodent species). This information should be very clearly articulated and discussed in the manuscript text itself, since it forms the basis for the experimental system used.*

The detailed culture conditions are now incorporated under “Cell culture” in the Materials and methods section. In particular, these cultures were kept under low-oxygen conditions (3% O_2_) after thawing to minimize selection for resistance to O_2_ toxicity. All of the preparations used for RNA and metabolite testing were similar in degree of expansion, all were at the earliest possible passage number from the original skin biopsy, and were expanded under low O_2_ conditions.

The cell lines were all exposed to the same culture conditions. We do not know if the conditions were "optimal" for rapid growth of cells from each species, but clearly the use of identical culture conditions is needed to avoid confounding effects due to species origin with possible effects of culture variations. We recently showed that cells from wild mice and other wild rodents are much less sensitive to O_2_ toxicity than cells of laboratory mice (Patrick et al., Aging 2016), but in any case we tried to minimize any such effects by use of 3% O_2_ and by doing all analyses on cells at early passage numbers.

*D) At what passage number were gene expression and metabolite profiles generated?*

See answer to point C) above and the revised Materials and methods section. Cryopreserved cells were prepared at passage 2 (counting the first transfer of confluent cells as passage 1); thawed aliquots of 10^6^ cells were then allowed to expand to 30 x 10^6^ cells, an increase of approximately 4 or 5 doublings.

*E) What is the in vitro proliferation rate of the different fibroblast isolates, and does this correlate with any of the gene expression profiles or longevity traits analyzed?*

We do not know the "in vitro proliferation" rate of our cell lines. This would vary in complex ways with seeding density, glucose and serum levels, degree of confluence at subculture, and days after seeding at which the proliferation was to be measured (lag phase, log expansion phase, or near-confluence, for example). It is possible that cell lines from different species might differ in some of these parameters, and that these differences could contribute to species differences in gene expression in a pattern that reflects longevity of the species tested, but evaluating this idea would be a complex undertaking. Nevertheless, the fibroblasts used for gene expression were collected at similar degree of confluency (with ~ 30 x 10^6^cells), so any variations in growth rate will be less likely to influence the gene expression pattern.

*2) It should also be discussed that* in vitro *cell culture conditions dramatically change chromatin architecture (Zhu et al., Cell 2013), and so gene expression patterns. The gene expression signature detected may simply reflect the ability of cells to differentially adjust to the* in vitro *conditions, and may contribute to the multiplex stress resistance. This caveat should be discussed in the text.*

We agree with the reviewers and have included this point in the Discussion. We used cells from the second/third passage for the RNA-seq and metabolite measurements in order to minimize in vitro culture artifacts, but it is still possible that some of the features reflect the differential responses to in vitro culture stress.

*3) It is not clear whether the common 9,389 gene orthologs that were reliably detected across the 15 species are comparable to the number of expressed genes detected by RNA-seq in fibroblasts in the 5 species with annotated genomes. The base line information on how many transcripts are detected in the 5 species should be available as they may provide a way to estimate the number of potentially missing orthologs, due to sequence divergence, some of which may be critical contributors to longevity through drastic functional alterations of gene products. The text should be clarified to address this point.*

We appreciate these comments and have provided the additional information in the main text and in [Supplementary-material SD1-data] “Mapping to Genomes”.

[Supplementary-material SD1-data] compares the results of mapping to the publicly available genomes and to our ortholog sets. 5 species (big brown bat, little brown bat, guinea pig, chinchilla, and deer mouse) have publicly available genomes in ENSEMBL and NCBI. For these species, we mapped the reads to the full genomes and the associated annotations, and counted the number of expressed genes (>10 counts).

The number of annotated genes in these species ranged between ~14,000 to ~16,000. Such variations are likely due to the different pipelines used for gene annotation. Among these annotated genes, ~10,000-11,000 had >10 counts (the same criteria for the expressed orthologs). This is similar to the 9389 orthologs we identified, indicating that our list of expressed orthologs is likely to have captured at least 80% of the expressed genes.

*4) It is not clear what the baseline gene set was for the enrichment analysis of the 827 top hits (Figure 3). Was it 9,389 genes or whole gene sets? And if the latter, which species? The data should be presented to show the enriched pathways in the top hits using the whole gene sets vs. 9,389 genes as baseline. This is because if DNA repair pathways are already enriched among the 9,389 genes, the conclusion that it is a longevity-associated molecular signature can be misleading. It is formerly possible that genes in these ancient pathways (such as DNA repair or glucose metabolism) may be more conserved sequence-wise and therefore more enriched among the orthologs, and thus among top hits. The text and data presentation should be clarified to address this point.*

We have clarified the pathway enrichment procedure in the Materials and methods section.

The background for pathway enrichment was based on the 16,816 unique protein-coding genes identified as Mouse Reference (i.e. Step 1 in Figure 2, after removing highly repetitive and highly similar sequences). The pathways were based on *Mus musculus*. We have now also included pathway enrichment statistics performed using only the 9389 expressed orthologs as background ([Supplementary-material SD2-data]). In both cases, the top enriched pathways remained statistically significant.

Regarding the comment “It is formerly possible that genes in these ancient pathways (such as DNA repair or glucose metabolism) may be more conserved sequence-wise and therefore more enriched among the orthologs, and thus among top hits”, we would like to discuss it in terms of the following aspects:

1) We agree with the reviewers that the genes in the DNA repair and glucose metabolism pathways are likely well conserved across the species. In fact, our additional analysis suggested that many of our ML-associated genes are “housekeeping genes” ([Supplementary-material SD2-data]). This is consistent with the notion that lifespan regulation across the species likely requires the co-ordination of a number of basic biological and cellular pathways. Since the longevity variation falls along a continuum across the different mammalian species, it is logical that the underlying evolution forces act on a set of pathways that are common (and conserved) across these species (i.e. the “public” pathways).

2) To the extent that our results are biased by sequence conservation (i.e. that we identified the DNA repair and glucose metabolism pathways only because of their high sequence conservation, and that we miss out many other important genes due to our ortholog identification strategy), we do not believe this is the case for the following reasons:

A) First, for those species with publicly available genomes, when we mapped the reads to the public genomes and used the public annotations, only ~10,000 to ~11,000 genes were expressed (based on a criterion of > 10 counts; same criterion in our ortholog filtering step), even though ~14,000 to ~16,000 genes were annotated ([Supplementary-material SD1-data]). This is mainly because the expression of many genes are tissue-specific, so they are simply not expressed in fibroblasts. In comparison, we identified 9389 expressed ortholog sets across all 15 species. Our 9389 orthologs should therefore represent at least 80% of all the expressed genes in fibroblasts of each species, and the correlation was between 0.93 to 0.98 for the read counts.

B) To address the issue of sequence conservation and divergence, we calculated the median DNA distance (using Kimura 2-parameters distance) for each ortholog set (Figure 2—figure supplement 1). In particular, among our 9389 expressed orthologs, there was no statistical difference in DNA distance between those orthologs that showed correlation with longevity and those orthologs that did not (Wilcoxon Rank Sum Test p value = 0.32; see Figure 6). In other words, the degree of sequence conservation or divergence is not linked to the degree of correlation with longevity. There are many other genes that are either more conserved or less conserved in sequence than the genes in DNA repair or glucose metabolism; but these other genes did not show up in the top hits because of their lack of expression correlation to longevity, not because of their sequence conservation or divergence per se.

Author response image 1.**DOI:**
http://dx.doi.org/10.7554/eLife.19130.017

*5) Subsection “Longevity trait variation across mammal”. The authors need to provide more information on how they calculated residuals from body mass. The reference to Ma et al. 2015b gives allometric equations, but no information is provided regarding where those equations come from. The description in Ma et al. (residual LS = LS/(a x Mass^b^)) implies that residuals were taken from a least-squares regression of log(LS) vs. log(Mass) with an intercept of log(a) and a slope of b. But this is not mentioned in the present manuscript, and must be inferred from the Ma et al. manuscript.*

We have now included an additional paragraph (“Life history data of the species”) under the Methods section to provide more information on the allometric equation. The ML equation, MLres = ML/(4.88×AW^0.153^), was based directly on the documentation of the AnAge database (http://genomics.senescence.info/help.html#anage). The relevant paragraph was quoted here:

“For mammals, also included is the maximum longevity (*tmax*) residual, expressed as a percentage of the expected maximum longevity calculated from the adult body size (*M*) and derived from the mammalian allometric equation: *tmax = M*^0.153^. This is useful to identify species that live longer than expected for their body size. Cetaceans were excluded because we have less confidence in their longevity records, obtained from studies in the wild often using indirect methods, than in those from other mammalian taxa.”

The FTM equation, FTMres = FTM/(78.1×AW^0.217^), was calculated by us, by linear regression using the FTM and body mass records of 1330 mammalian species in the AnAge database.

If we had used the same procedure to calculate the MLres equation, we would obtain MLres = ML/(4.36×AW^0.158^), which is comparable to the AnAge documentation.

*If a least-squares regression approach was used, one important assumption is that the residuals are normally distributed. The data for Figure 1 (MLres and FTMres) do not look normally distributed. If they are not, a general linear model with an appropriate link function must be applied.*

We had tested the normalcy of the data using Shapiro test. The results were as follows (if p-value < 0.05, then the data is not normally distributed):

Trait (on log_10_ scale)Shapiro test p-valueML0.8401FTM0.0753MLres0.7969FTMres0.1258

Therefore, none of the longevity traits violated normalcy assumption.

Furthermore, we also tested the normalcy of the gene expression data and metabolite data. Using Shapiro test, 89% of the genes and 90% of the metabolites did not violate normalcy assumption on log_10_ scale.

*6) The authors use maximum lifespan as a metric of aging, and thus should add to the text of their manuscript a discussion of the concerns that have been raised regarding this statistic (e.g. Moorad et al. 2012 Aging Cell). In particular, it does not provide a measure of aging per se, is quite sensitive to small sample size, and is not in itself under direct selection.*

Thank you. We have included this point in the Discussion section. Additional parameters such as mean adult lifespan and 90^th^ percentile of longevity (Moorad et al., 2012) should be used in place of ML in the long run, although at the moment such records are available for only a limited number of species.

*7) Analyses of primate and bird species (subsection “Validation of amino acid patterns in primate and bird fibroblasts”) does not appear to be size corrected. Numerous studies have shown strong size effects of life span not only in primates, but also in birds. Therefore, size effects should be addressed here as well.*

We have now expanded Table 2 to include the body-mass adjusted lifespan correlation. We would like to point out that:

A) Removal of body mass weakened the observed correlation (Table 2). This is likely due to the inherent strong correlation between maximum lifespan and body mass. Indeed, the same weakening is also observed among the rodent fibroblasts ([Supplementary-material SD3-data]), indicating the two datasets are in fact consistent.

B) The same weakening of correlation is also observed in our previous studies (Ma et al., 2015 Cell Metabolism and Ma et al., 2015 Cell Reports). This is in fact expected, given the strong correlation between maximum lifespan and body mass. It is highly debatable whether one could completely disentangle these 2 traits, since they are likely under the same natural selection force.

C) Whether the body-mass should be removed in cross-species longevity studies is an area where experts in comparative biology hold widely divergent views. Some (e.g. John Speakman) think that "uncorrected" data are not interpretable; another set, equally experienced (e.g. Tony Hulbert), thinks that an attempt to "correct" for mass will create a major risk for missing key associations, since mass and lifespan are so strongly confounded across species. Others suggest that one should report both unadjusted and adjusted values so that readers can see whichever set of results they feel is most valid.

D) Such associations, whether or not corrected for mass, are only a first step towards more definite, causal, hypotheses. Some remain significant under both ML and MLres, while others do not. Nevertheless, they can all be potential candidates for further experimental verification.

*8) Subsection “Evaluation of amino acid levels in bird and primate fibroblast cell lines”. For the bird/primate analysis, the authors state, "when two or more […] features were annotated as corresponding to the same amino acid, we tabulated the degree of association from the feature most strongly correlated with lifespan among the species studied." This approach will necessarily bias the result in favor of suggesting a stronger relationship between lifespan and metabolite levels than might be true, and this caveat must be mentioned.*

This selection reflects, in part, uncertainties in the annotation of specific features in the untargeted metabolomics protocol used for the bird and primate species. In most cases where two or more of the annotated features were thought to represent the same amino acid, there was very good agreement. The table below gives the key information on this point. It lists each amino acid for which there was more than one annotated feature, and gives the p-values for the relationship with species lifespan for each such feature.

**Amino acid****p-values**Arginine0.03, 0.1, 0.2Glutamic acid0.01, 0.2Tryptophan0.001, 0.001, 0.01Methionine0.03, 0.1Phenylalanine0.001, 0.003, 0.01Proline0.001, 0.001Valine0.001, 0.001

In four cases (tryptophan, phenylalanine, proline, valine), all such features show a significant association with lifespan. For the other three, the features that do not meet the traditional p = 0.05 criterion have two-sided p-values between 0.05 and 0.2, with the same sign of the slope coefficient.

We agree with the reviewer that our decision to present the feature with the best fit to the MLS regression could in principle create bias and thus increase Type I error, but in this case there are several reasons for confidence: the close fit between each of the nominal "replicate" features within the bird/primate data set, the good agreement across each of the 10 amino acids evaluated, and the excellent agreement to the relationship seen for the rodent cell lines evaluated with an entirely different method.